# Solanum americanum genome-assisted discovery of immune receptors that detect potato late blight pathogen effectors

Xiao Lin [1,2,12] ✉, Yuxin Jia[3,4,12], Robert Heal[1], Maxim Prokchorchik [5,11], Maria Sindalovskaya[1], Andrea Olave-Achury [1], Moffat Makechemu[1], Sebastian Fairhead[1], Azka Noureen[1], Jung Heo[6], Kamil Witek[1], Matthew Smoker[1], Jodie Taylor[1], Ram-Krishna Shrestha [1], Yoonyoung Lee[5], Chunzhi Zhang [3], Soon Ju Park [6,7], Kee Hoon Sohn [5,8,9,10] ✉, Sanwen Huang [3] ✉ & Jonathan D. G. Jones [1] ✉

Potato (*Solanum tuberosum*) and tomato (*Solanum lycopersicon*) crops suffer severe losses to late blight caused by the oomycete pathogen *Phytophthora infestans*. *Solanum americanum*, a relative of potato and tomato, is globally distributed and most accessions are highly blight resistant. We generated high-quality reference genomes of four *S. americanum* accessions, resequenced 52 accessions, and defined a pan-NLRome of *S. americanum* immune receptor genes. We further screened for variation in recognition of 315 *P. infestans* RXLR effectors in 52 *S. americanum* accessions. Using these genomic and phenotypic data, we cloned three NLR-encoding genes, *Rpi-amr4*, *R02860* and *R04373*, that recognize cognate *P. infestans* RXLR effectors *PITG_22825* (*AVRamr4*), *PITG_02860* and *PITG_04373*. These genomic resources and methodologies will support efforts to engineer potatoes with durable late blight resistance and can be applied to diseases of other crops.

Potato is one of the most consumed nongrain crops worldwide. However, pests and diseases reduce global yields by ~17% (ref. 1). Potato late blight, which is caused by the oomycete pathogen *Phytophthora infestans*[2], triggered the Irish famine in the 1840s and is still the most damaging disease for global potato production[1].

Plant immunity depends on pathogen recognition by both cell-surface pattern recognition receptors (PRRs) and intracellular immune receptors. Many *R* genes against *P. infestans* (*Rpi* genes) were cloned from wild relatives of potato species, such as *R2*, *R3a*, *R8*, *Rpi-blb1*, *Rpi-blb2* and *Rpi-vnt1* from *Solanum demissum*, *Solanum*

[1]The Sainsbury Laboratory, University of East Anglia, Norwich Research Park, Norwich, UK. [2]State Key Laboratory of Plant Genomics, Institute of Microbiology, Chinese Academy of Sciences, Beijing, China. [3]Shenzhen Branch, Guangdong Laboratory of Lingnan Modern Agriculture, Genome Analysis Laboratory of the Ministry of Agriculture and Rural Area, Agricultural Genomics Institute at Shenzhen, Chinese Academy of Agricultural Sciences, Shenzhen, China. [4]Key Laboratory for Potato Biology of Yunnan Province, The CAAS-YNNU-YINMORE Joint Academy of Potato Science, Yunnan Normal University, Kunming, China. [5]Department of Life Sciences, Pohang University of Science and Technology, Pohang, Republic of Korea. [6]Department of Biological Science and Institute of Basic Science, Wonkwang University, Iksan, Republic of Korea. [7]Division of Applied Life Sciences and Plant Molecular Biology and Biotechnology Research Center (PMBBRC), Gyeongsang National University, Jinju, Republic of Korea. [8]School of Interdisciplinary Bioscience and Bioengineering, Pohang University of Science and Technology, Pohang, Republic of Korea. [9]Department of Agricultural Biotechnology, Seoul National University, Seoul, Republic of Korea. [10]Plant Immunity Research Center, Seoul National University, Seoul, Republic of Korea. [11]Present address: Plant Pathology Group, The Institute of Crop Science and Resource Conservation, University of Bonn, Bonn, Germany. [12]These authors contributed equally: Xiao Lin, Yuxin Jia. ✉e-mail: linx@im.ac.cn; keehoon.sohn@snu.ac.kr; huangsanwen@caas.cn; jonathan.jones@tsl.ac.uk

*bulbocastanum* and *Solanum venturii*[3–10]. However, most cloned *Rpi* genes have been overcome by the fast-evolving pathogen.

*P. infestans* effectors carry a signal peptide and an RXLR-EER motif (where X represents any amino acid). In the *P. infestans* reference genome (strain T30-4), 563 RXLR effectors were predicted, enabling screens for recognition of these effectors ('effectoromics') in various plants[11,12].

Reference genome sequences of potato, tomato, eggplant and pepper have been determined[13–16]. Phased, chromosome-level genome assemblies of heterozygous diploid and tetraploid potatoes are also available[17–19]. Pan-genome studies of crop plants including potato have also emerged that shed light on the extensive genetic variation in these species[20–23]. Sequence capture methods have been developed to sequence plant *NLR* (RenSeq) and *PRR* (RLP/KSeq) gene repertoires that reduce the genomic complexity and sequencing costs[24–26]. These methods have led to many important applications, such as AgRenSeq, and defining the pan-NLRome of *Arabidopsis*[27,28].

Diploid *Solanum americanum* is highly resistant to late blight. Previously, our group cloned *Rpi-amr1* and *Rpi-amr3* from several resistant *S. americanum* accessions along with their cognate effectors AVRamr1 and AVRamr3 (refs. 25,29–31).

Here, we sequenced and assembled four high-quality genomes of *S. americanum*, resequenced 52 accessions, and defined the pan-NLRome of *S. americanum*. We also screened 315 *P. infestans* RXLR effectors in 52 *S. americanum* accessions. These genomic resources and functional data led to the rapid identification of three new NLR-encoding genes, *Rpi-amr4*, *RO2860* and *RO4373*, that are responsible for PITG_22825 (AVRamr4), PITG_02860 and PITG_04373 recognition, respectively. This study unveils an effector-triggered immunity (ETI) interaction landscape between *S. americanum* and *P. infestans* that will enable us to clone more *Rpi* genes from the gene pool of wild *Solanum* species and deepen our knowledge of late blight resistance in wild relatives of potato. Potato genome design driven by potato genomics that takes advantage of novel plant breeding technologies[32] will help to develop better potato varieties with durable late blight resistance.

## Results

### Genome assembly and gene model prediction of *S. americanum*

*S. americanum* is a globally distributed Solanaceae species that is resistant to many pathogens, including *P. infestans* and *Ralstonia solanacearum*[25,29,33]. Four *S. americanum* accessions SP1102, SP2271, SP2273 and SP2275 were selected for sequencing based on their variation in resistance to late blight (Supplementary Fig. 1a,b). We generated PacBio high-fidelity, Oxford Nanopore and Illumina paired-end reads and assembled the genomes of SP1102, SP2271, SP2273 and SP2275 into contigs (Supplementary Note 1). We also generated Hi-C data for SP1102, SP2271 and SP2273, and anchored the contigs into 12 pseudomolecules (Supplementary Note 1 and Supplementary Figs. 2 and 3). The completeness of these assemblies was estimated to be ~98.4% (single-copy and duplicated) by BUSCO, which indicates the high quality of genome assembly (Supplementary Fig. 4a). To annotate gene models, we applied EVidenceModeler or GeMoMa pipelines to integrate the ab initio prediction, homology-based annotation and transcriptome evidence for SP1102/SP2271 or SP2273/SP2275. In summary, we predicted an average of 34,193 gene models with an average of 98.1% BUSCO evaluation (single-copy and duplicated) for each *S. americanum* genome (Supplementary Fig. 4b and Table 1).

### Genome evolution of *S. americanum*

To investigate the evolution of *S. americanum* genomes, we clustered the representative protein sequences from 15 genomes, comprising the genomes from four *S. americanum* accessions, four potato accessions, three tomato accessions, four additional Solanaceae species and an outgroup species of *Arabidopsis thaliana*, into 33,115 orthogroups, from which we further identified 1,363 single-copy orthogroups. The species

**Table 1 | Summary of genome assembly and annotation for *S. americanum***

| Genome features | SP1102 | SP2271 | SP2273 | SP2275 |
|---|---|---|---|---|
| Sequencing method | CCS, Hi-C | CCS, Hi-C | ONT, Illumina, Hi-C | ONT, Illumina |
| Estimated genome size | 1.15 Gb | 1.21 Gb | 1.31 Gb | 1.21 Gb |
| Heterozygosity | 0.34% | 0.35% | 0.05% | 0.06% |
| Total assembly | 1.07 Gb | 1.13 Gb | 1.02 Gb | 1.02 Gb |
| Contigs | 299 | 568 | 294 | 550 |
| Contig N50 | 82.9 Mb | 55.2 Mb | 10.3 Mb | 4.8 Mb |
| Contig L50 | 7 | 8 | 30 | 66 |
| Chromosomes | 12 | 12 | 12 | – |
| Anchor rate | 98.43% | 97.60% | 99.31% | – |
| BUSCO for assembly | 98.40% | 98.30% | 98.40% | 98.40% |
| Gene models | 35,654 | 36,073 | 31,976 | 33,051 |
| Transcripts | 35,654 | 36,073 | 42,391 | 44,027 |
| BUSCO for annotation | 97.70% | 97.90% | 98.60% | 98.00% |

tree topology suggests that *S. americanum* is a sister species to the common ancestor of potato and tomato and diverged ~14.1 million years ago (Ma; 95% highest posterior density interval, 11.7–17.2 Ma; Fig. 1a), which is consistent with a former report based on plastid sequences[34].

Chromosome rearrangement (CR) is an important evolutionary process[35]. The reference-grade genome assemblies enabled us to explore *S. americanum* chromosome evolution. We observed 45 large CRs (>1 Mb in size), comprising 26 inversion and 19 inter-chromosome translocation events, between the *S. americanum* and potato genomes (Fig. 1b and Supplementary Fig. 5). In contrast, 67 large CRs (30 inversions and 37 inter-chromosome translocations) were found between *S. americanum* and eggplant (Fig. 1b). Notably, CRs were not evenly distributed across the genome. No CR was identified on chromosome 2 between *S. americanum* and potato, while 11 CRs occurred on chromosome 11.

### Structural variation between *S. americanum* genomes

Structural variations (SVs), including insertions, deletions, duplications, inversions and translocations, cause and maintain phenotypic diversity[36]. The chromosome assemblies of three *S. americanum* genomes enabled the analysis of large SVs (>1 Mb in size). Using SP1102 as the reference, we identified 56 large SVs in SP2271 (Supplementary Fig. 6a), impacting ~256 Mb of the reference genome. However, only 14 large SVs were identified in SP2273, covering ~54 Mb of the reference genome (Supplementary Fig. 6b). Most of the SVs reside in single contigs and are supported by the Hi-C interaction map, suggesting the high reliability of SV identification (Supplementary Fig. 6c and Supplementary Table 1). The large differences in SV numbers among *S. americanum* genomes shed light on their complex evolutionary history. We further characterized the small SVs (40 bp–1 Mb in size) among *S. americanum* genomes and found that SVs might contribute to the differential expression of 1,084 genes between SP1102 and SP2271 leaves (Supplementary Note 2 and Supplementary Figs. 7 and 8).

### Defining the *S. americanum* pan-NLR repertoire

To understand *NLR* gene diversity, a phylogenetic tree was generated using the NB-ARC domain of the NLR proteins from SP1102 (Fig. 2a) and the position of these *NLR* genes in the SP1102 genome was visualized in the physical map (Fig. 2b). We found that 71% of SP1102 *NLR* genes were in clusters and the rest were singletons (Fig. 2c). Due to the complexity

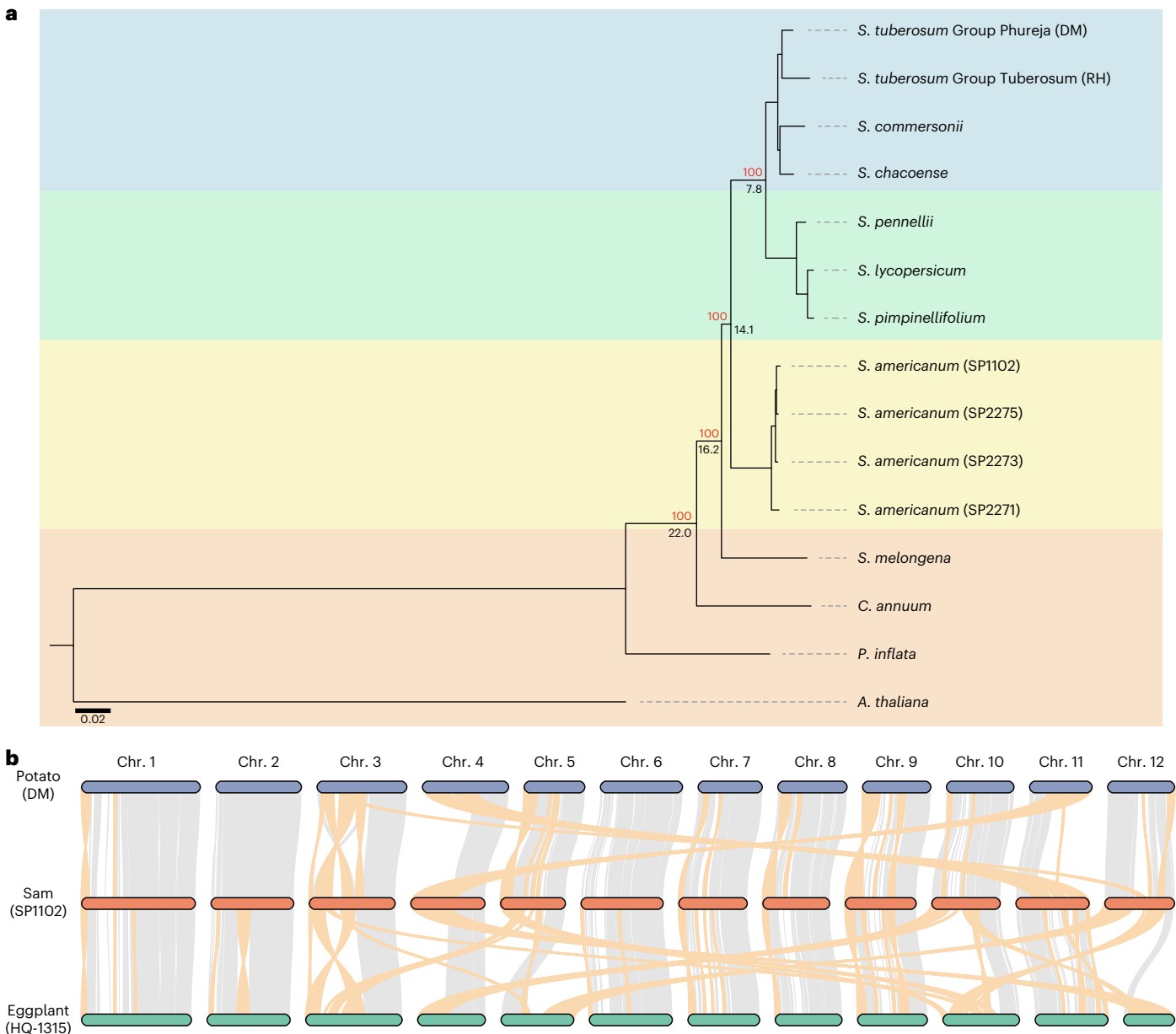

**Fig. 1 | Genome evolution of *Solanum americanum*. a**, Phylogenetic relationship of *S. americanum* and neighboring species. The red number indicates the bootstrap of each node. The black number denotes the estimated divergence time (million years ago). The scale bar represents the number of amino acid substitutions per site. **b**, Genome synteny of *S. americanum*, potato and eggplant. Ribbons between chromosomes show syntenic regions. Large chromosome rearrangements (>1 Mb in size) are marked in orange.

of *NLR* gene clusters, most automatic annotation pipelines produce incorrect gene models[37]. To generate better models of the *NLR* genes, we manually annotated 528, 579 and 524 *NLR* genes from SP1102, SP2271 and SP2273 genomes by incorporating NLR-annotator results and cDNA sequence data (Fig. 2d). Next, presence/absence (P/A) polymorphisms of *NLR* genes among *S. americanum* accessions were compared (Fig. 2a). Further, a pan-NLRome was built, which suggests that the accessions in our research are representative of the *S. americanum NLR* repertoire (Supplementary Note 3 and Supplementary Fig. 9).

We also inspected the expression level of SP1102 *NLR* genes by re-analyzing RenSeq cDNA data[25]. The transcripts per million (TPM) values of *NLR* genes were visualized with a heatmap (Fig. 2a). Many well-known *NLR* genes were relatively highly expressed, such as the sensor NLR *Rpi-amr3*, and helper NLRs *ADR1*, *NRG1*, *NRC4a*, *NRC2* and *NRC3* (Fig. 2f and Supplementary Table 2). Many Solanaceae coiled-coil

NLRs (CC-NLRs, or CNLs) require helper NLR NRCs that are phylogenetically related and we found that about 50% of the *S. americanum* NLRs lie within the NRC superclade[38] (Fig. 2a). To investigate the NRC family, we generated a phylogenetic tree for the *NRC* genes. We found *NRC1*, *NRC2*, *NRC3*, *NRC4a* and *NRC6* homologs, and two *NRC5a* homologs in the *S. americanum* genome (Fig. 2b, e). Interestingly, *NRC4b* genes (seven homologs) have expanded in the *S. americanum* genome compared to *Nicotiana benthamiana* (two homologs) (Fig. 2e). Previously, we reported that Rpi-amr3 and Rpi-amr1 require NbNRC2/NbNRC3/NbNRC4 and NbNRC2/NbNRC3 in *N. benthamiana*, respectively[29,30]. However, *NRC1* is missing in *N. benthamiana*[39]. To test whether *NRC1* from *S. americanum* can support Rpi-amr1/Rpi-amr3 function, we cloned the *SaNRC1* from SP1102 and showed that SaNRC1 enables Rpi-amr3 but not Rpi-amr1 function in *N. benthamiana nrc2/mc3/mc4* knockout plants (Supplementary Fig. 10). This result indicates

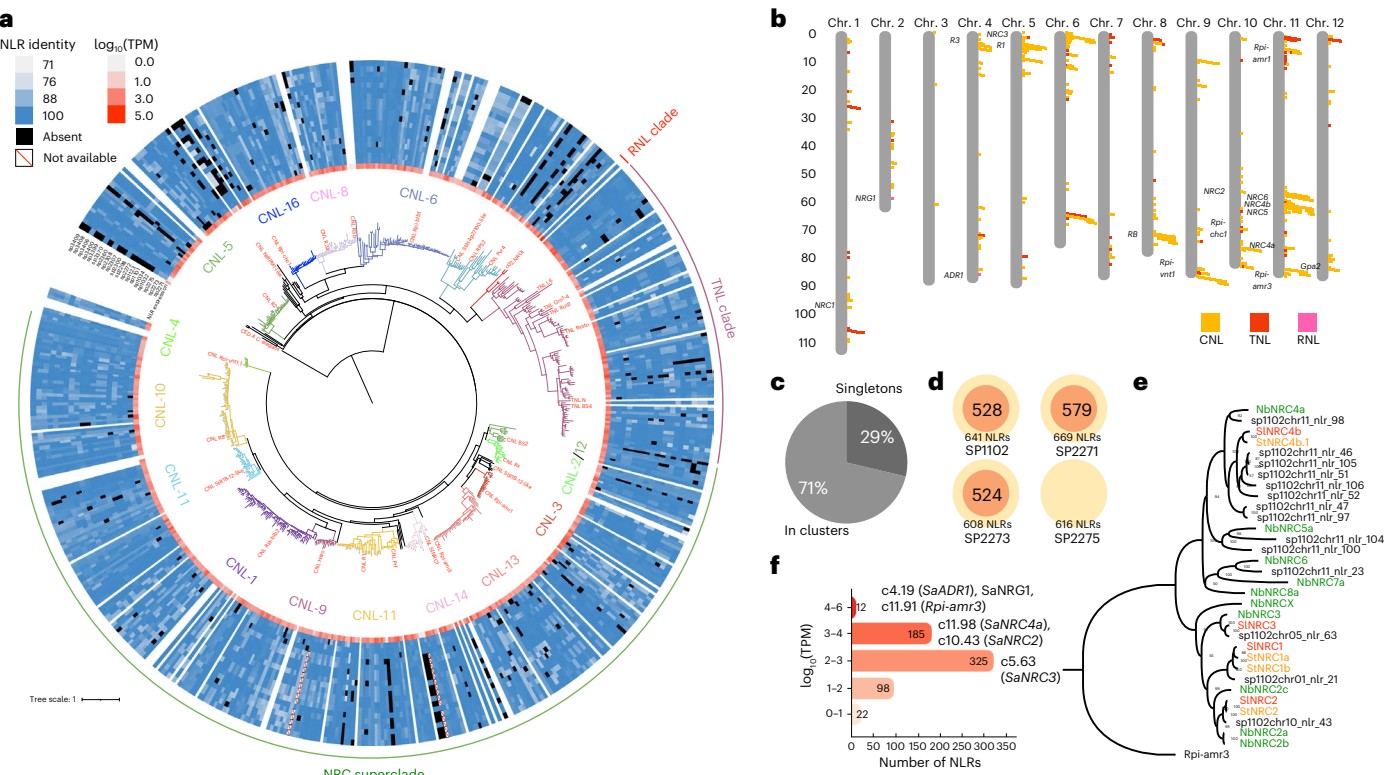

**Fig. 2 | Pan-NLRome of *S. americanum*. a**, The NB-ARC domains of *S. americanum* SP1102 were predicted by NLR-annotator and used to generate a maximum-likelihood tree using IQ-TREE with the JTT + F + R9 model. Known NLR proteins from Solanaceae species were included (highlighted in red). The NLRs are classified into different groups based on a previous report[29]. The RNL, TNL and NRC superclade are shown. CED-4 from *Caenorhabditis elegans* was used as the outgroup. The expression profile is shown by a heatmap (white to red) based on the cDNA RenSeq data of SP1102. The P/A polymorphism of NLRs from the three other *S. americanum* genomes and SMRT RenSeq assemblies of 16 additional accessions are shown by the heatmap (white to blue). The accession order from top to bottom is SP3409, SP3408, SP3406, SP3400, SP3399, SP3370, SP2360, SP2308, SP2307, SP2300, SP2298, SP2272, SP1123, SP1101, SP1034, SP1032, SP2275, SP2273 and SP2271. The absent NLRs are shown by black blocks. The scale bar represents the number of amino acid substitutions per site. **b**, The physical map of *NLR* genes in the SP1102 genome. CNLs are shown by yellow blocks, TNLs are shown by red blocks and RNLs are shown by pink blocks. Some functionally characterized *NLR* clusters are noted on this map. Some previously reported *NLR* clusters (*NRC1, NRG1, R3, ADR1, NRC3, R1, RB, Rpi-vnt1, NRC2, Rpi-chc1, Rpi-amr1, NRC6, NRC4b, NRC5, NRC4a, Rpi-amr3, Gpa2*) are shown in the physical map. (**c**) The proportion of *NLR* singletons and *NLRs* in clusters. **d**, Number of manually curated *NLR* genes (red circle), and the number of *NLR* genes predicted by NLR-annotator (yellow circle). Manual curation of *NLR* genes from SP2275 was not performed. **e**, Phylogeny of the NRC helper NLR family. The NRC homologs from potato, tomato and *N. benthamiana* are marked in orange, red and green, respectively. The NRC proteins from *S. americanum* are in black. The number indicates the bootstrap of each node. The scale bar represents the number of amino acid substitutions per site. **f**, The log$_{10}$ transformed TPM values for *NLR* genes are classified into five groups, and some homologs of known *R* genes are noted. The NLR IDs are shown in Supplementary Table 2.

that distinct members in different plant species might enable NRC functions.

The *S. americanum* genome and pan-NLRome also enabled us to investigate the diversity of *NLR* genes. We found that sequence diversity in *NLR* regions was significantly higher than in non-NLR regions (Supplementary Fig. 11a), consistent with a previous report[37]. In addition, we found extensive sequence diversity and copy number variation within *NLR* clusters. For example, the *Rpi-amr3* locus varied greatly among the *S. americanum* genomes (Supplementary Fig. 11b), showing that high-quality genomes are required for reliable NLRome annotation.

In summary, we generated a pan-NLRome of 20 *S. americanum* accessions and manually annotated the *NLR* genes from three reference genomes. This resource is important for the investigation of *NLR* gene evolution and facilitates functional studies of ETI in *S. americanum* and other Solanaceae species.

## The ETI landscape of the *S. americanum* and *P. infestans* interaction

There are 563 predicted RXLR effectors in the T30-4 *P. infestans* reference genome. In this study, we showed that there are ~550 *NLR* genes

in *S. americanum* reference genomes (Fig. 2d). To reveal one-to-one effector-receptor interactions and clone more immune receptors, we screened ~315 RXLR effectors on 52 *S. americanum* accessions (Fig. 3). Based on cDNA PenSeq data, all these RXLR effectors are expressed during colonization of a susceptible potato leaf[31]. We found that five effectors triggered hypersensitive response (HR) on most *S. americanum* accessions (≥ 50), including effectors from the AVRblb2 family, while 185 effectors did not trigger HR in any *S. americanum* accessions, 71 effectors were recognized by fewer than five *S. americanum* accessions and 54 effectors showed differential recognition by different *S. americanum* accessions. AVRamr1 (36/52) and AVRamr3 (43/52) were also widely recognized by different *S. americanum* accessions (Fig. 3). The four reference accessions SP2271, SP2275, SP1102 and SP2273 could recognize 25, 18, 30 and 30 RXLR effectors, respectively, of which 5, 3, 7 and 9 effectors were specifically recognized in each accession (Supplementary Fig. 12). Notably, accession SP2271 was susceptible to *P. infestans* in the detached leaf assay (DLA), but susceptibility was age-dependent (Supplementary Fig. 1b), and this accession is resistant to late blight in the field. As expected, SP2271 did not recognize AVRamr1 and AVRamr3. We found premature stop codons

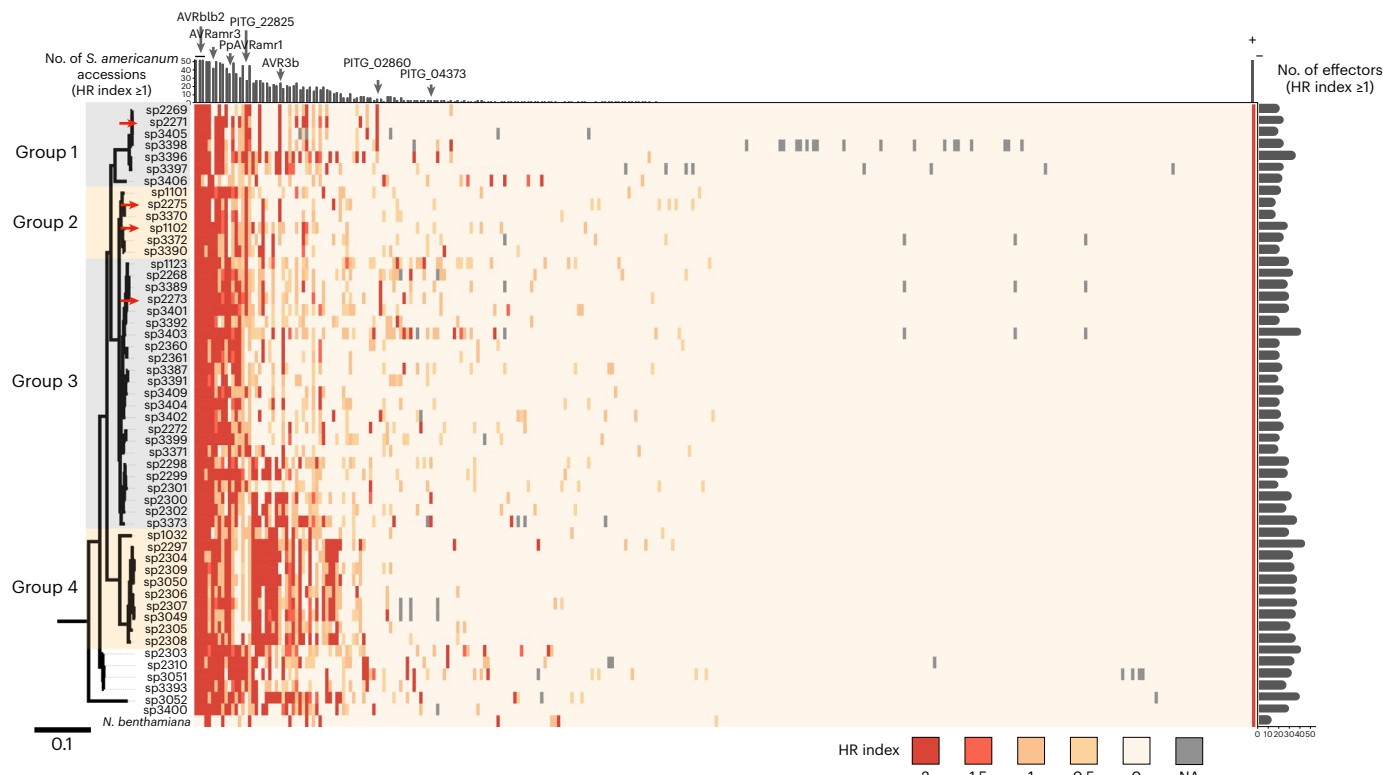

**Fig. 3 | ETI landscape of *S. americanum* and *P. infestans*.** A total of 315 RXLR effectors were transiently expressed in 52 *S. americanum* accessions. The HR index (2, strong HR; 1, weak HR; 0, no HR, NA, not available) was used for the heatmap. These effectors were screened on *N. benthamiana*[30], and their recognitions are included in this heatmap. Empty PVX vector was used as negative control, and co-expression of Rpi-amr3–HisFlag and AVRamr3–GFP was used as positive control. The *S. americanum* accessions were ordered based on the phylogenetic tree; SP3400 is not included in this tree. The scale bar represents the number of amino acid substitutions per site. The *S. americanum* accessions were classified into four groups (gray or yellow shading). The four reference accessions are marked by red arrows. The effectors were ordered based on the total HR index. For each effector, the numbers of responsive *S. americanum* accessions is visualized by a bar chart on the top of the heatmap. For each *S. americanum* accession, the numbers of recognized effectors is visualized by a bar chart on the right of the heatmap. Some RXLR effectors previously characterized or mentioned in this study are indicated by gray arrows.

in both *Rpi-amr1* and *Rpi-amr3* homologs from SP2271. Intriguingly, 25 RXLR effectors triggered HR in SP2271. These RXLR effector recognitions might contribute to the age-dependent resistance and field resistance of SP2271 to late blight. Taken together, these results reveal the ETI landscape of *S. americanum* against the late blight pathogen.

**Resequencing of *S. americanum* accessions**

To investigate genetic diversity in the *S. americanum* accessions in our collection, we performed PCR-free, 150 bp paired-end sequencing for 52 *S. americanum* accessions at 10× coverage. We constructed a phylogenetic tree using all genic SNPs, and eggplant, potato, and tomato were used as outgroups (Fig. 3 and Supplementary Fig. 13a). Structural and inbreeding coefficient values were also analyzed (Supplementary Fig. 13b, c). The 52 accessions can be assigned into four groups (Supplementary Fig. 13a). All six accessions in group 1 lacked *Rpi-amr1* and *Rpi-amr3* based on the effectoromics screening (Fig. 3 and Supplementary Table 3). SP2275 and SP1102 were in group 2 and SP2273 was in group 3. No reference genome was available for group 4, but we generated SMRT-RenSeq assemblies for several of these accessions. Surprisingly, four accessions (SP2303, SP2310, SP3393 and SP3051) were not closely related to other groups and are highly heterozygous (Supplementary Fig. 13c), suggesting that they may be polyploid species like *Solanum nigrum*. Two other accessions, SP3052 and SP3376, were also not closely related to the four *S. americanum* groups and might belong to another Solanaceae species. These resequencing data could be used for genome-wide association studies (GWAS) and molecular marker development.

**Cloning of *Rpi-amr4* by GWAS and linkage analysis**

During effectoromics screening, we found an effector, PITG_22825, that triggered HR in 28 of 52 *S. americanum* accessions (Fig. 3 and Supplementary Table 3), including SP1102 and SP2271 but not SP2298 (Fig. 4a). PITG_22825 is an RXLR effector with a signal peptide and RQLR and EER motifs followed by the effector domain (Fig. 4a). This effector had not received attention before our cDNA PenSeq study[31]. To map the gene conferring its recognition, a GWAS analysis was performed, and a strong signal was identified in an *NLR* singleton located on chromosome 01 of SP1102 (Figs. 2b and 4b). This gene encodes a CNL that belongs to the CNL-13 *Rpi-amr3* phylogenetic clade (Fig. 2a), although the *Rpi-amr3* gene cluster locates on chromosome 11 (Fig. 2b). This indicates that the candidate gene on chromosome 1 might have translocated from the *Rpi-amr3* locus on chromosome 11, which probably explains another weaker GWAS signal in the *Rpi-amr3* cluster of chromosome 11 (Fig. 4b). Based on cDNA RenSeq data from SP1102, the corresponding *NLR* gene carries an extra exon compared to *Rpi-amr3* (Fig. 4b). To verify this GWAS signal, we performed a bulked segregant analysis and resistance gene enrichment sequencing (BSA-RenSeq) in a segregating F₂ population of SP2271 x SP2298 (Supplementary Fig. 14). The PITG_22825 responsive gene from SP2271 was mapped to the same position on chromosome 1 in both the SP2271 and SP1102 reference genomes.

To test gene function, the ORFs of the candidate genes from SP2271 and SP1102 were PCR amplified and cloned into an over-expression binary vector with the 35S promoter and *Ocs* terminator and the resulting constructs were then transformed into *Agrobacterium*

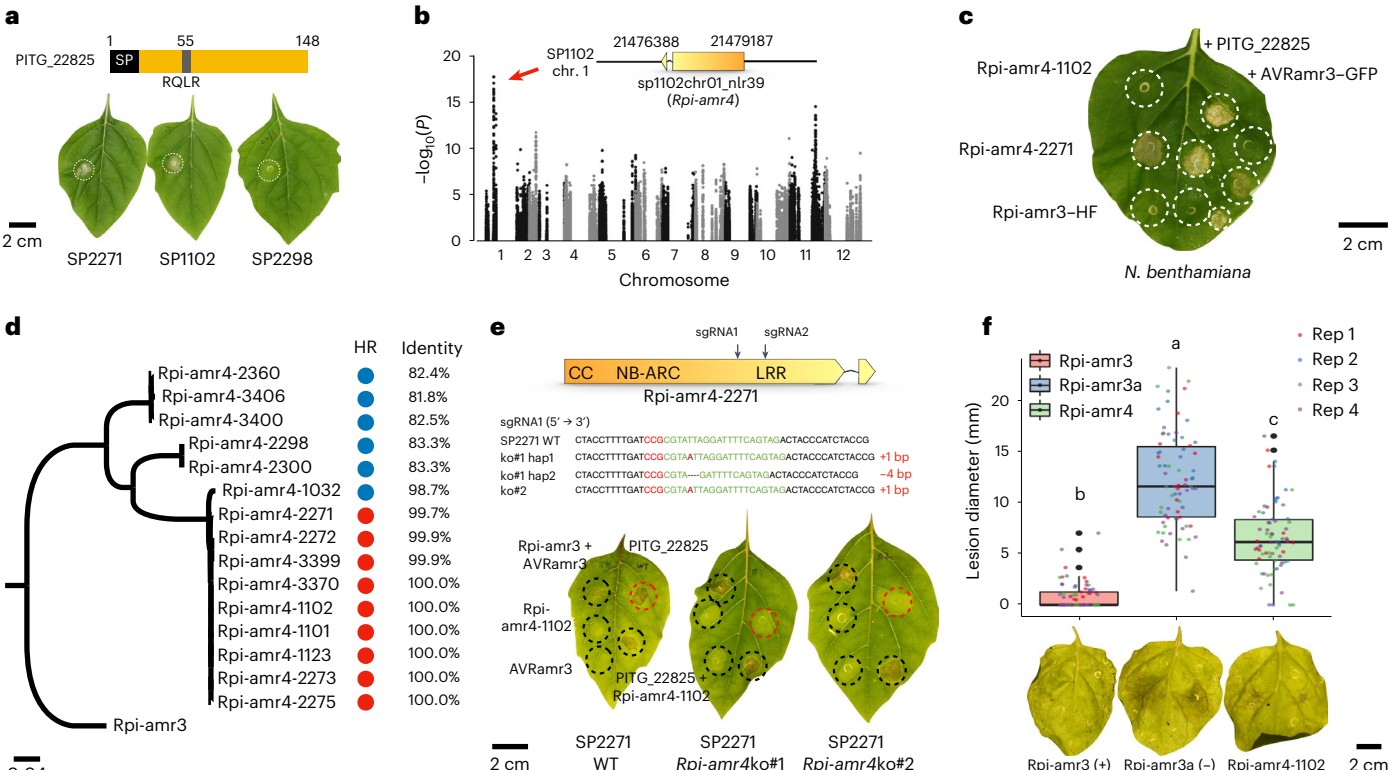

**Fig. 4 | Identification and characterization of *Rpi-amr4* that recognizes PITG_22825. a**, PITG_22825 is an RXLR effector. *35S::PITG_22825* triggers cell death on *S. americanum* SP2271 and SP1102 leaves, but not SP2298 leaves. **b**, Manhattan plot of the GWAS of PITG_22825 recognition. The SNPs associated with PITG_22825 recognition is located in an *NLR* singleton sp1102chr01_nlr39; (red arrow). **c**, HR assay of candidate genes. *Rpi-amr4-1102* and *Rpi-amr4-2271* were expressed alone or co-expressed with either *35S::PITG_22825* or *35S::AVRamr3–GFP* constructs in *N. benthamiana* leaves. Rpi-amr4-2271 is auto-active in *N. benthamiana*, but when co-expressed with PITG_22825, the HR was stronger. Rpi-amr4-1102 specifically recognizes PITG_22825. Rpi-amr3 was used as control. $OD_{600} = 0.5$. Four leaves from two plants were used for each experiment and three biological replicates were performed with the same results. HF, HisFlag. **d**, Phylogeny of Rpi-amr4 homologs in different *S. americanum* accessions. Rpi-amr3 was used as an outgroup. PITG_22825-mediated HR is shown by red (HR) or blue (no HR) circles. Percent identity of the amino acid sequence relative to Rpi-amr4-1102 is shown. **a,c,d–f**, The scale bars represent the number of amino acid substitutions per site. **e**, *Rpi-amr4*-knockout lines

lose the capacity for PITG_22825 recognition. Two sgRNAs (black arrows) were designed on *Rpi-amr4-2271*. The genotype of the two knockout lines is shown. Both lines failed to recognize PITG_22825, but HR could be complemented when co-expressing PITG_22825 with *Rpi-amr4-1102*. Wild-type (WT) SP2271 plants were used as control plants. *Rpi-amr3* and AVRamr3 were used as positive control. $OD_{600} = 0.5$. **f**, DLA with *35S::Rpi-amr4-1102*. *35S::Rpi-amr4-1102* (green), *Rpi-amr3* (positive control, red) and *Rpi-amr3a* (a non-functional *Rpi-amr3* paralog, negative control, blue) were transiently expressed in *N. benthamiana*, $OD_{600} = 0.5$. Zoospores from *P. infestans* strain T30-4 were used to inoculate the leaves 1 day post-infiltration (dpi). Lesion sizes were measured at 6 dpi. Four biological replicates were performed, and all data points (74 data points per treatment) were visualized as a box-and-whisker plot. Center line, median; box limits, upper and lower quartiles. The whiskers (top and bottom) comprise values within 1.5 times the interquartile range (IQR). The outliers are indicated by black dots. Statistical differences were analyzed by one-way ANOVA with Tukey's HSD test ($P < 0.001$) and were indicated by the lower-case letters. Representative leaves are shown.

*tumefaciens*. The candidate genes were expressed in *N. benthamiana* alone or co-expressed with PITG_22825 or AVRamr3-GFP as the negative control. Rpi-amr3-HisFlag was used as a positive control. We found that the SP2271 allele (*Rpi-amr4-2271* hereafter) was auto-active in *N. benthamiana*, but when it was co-expressed with PITG_22825 the HR was faster and stronger compared to the control (Fig. 4c). In contrast, the SP1102 allele (*Rpi-amr4-1102* hereafter) was not auto-active in *N. benthamiana*. HR was triggered when *Rpi-amr4-1102* was co-expressed with PITG_22825, but not AVRamr3–GFP (Fig. 4c). There are only three amino-acid differences between the proteins encoded by *Rpi-amr4-1102* and *Rpi-amr4-2271*, and these differences might cause the auto-activity of the SP2271 allele (Supplementary Fig. 15). We also found that *Rpi-amr4* was conserved in the PITG_22825 responsive accessions (Fig. 4d). To verify the function of *Rpi-amr4*, we generated *Rpi-amr4*-knockout SP2271 lines by CRISPR-Cas9. In total, 16 CRISPR-Cas9 knockout lines were generated (Supplementary Table 4) and 2 lines are shown in Fig. 4e. Wild-type SP2271 could recognize PITG_22825, but the *Rpi-amr4*-knockout lines could not.

The HR phenotype could be complemented when *Rpi-amr4-1102* was co-expressed with PITG_22825 in the knockout lines (Fig. 4e). Therefore, we conclude that *Rpi-amr4* encodes the PITG_22825-recognizing immune receptor and that *PITG_22825* is *Avramr4*.

To test whether *Rpi-amr4* confers late blight resistance, we transiently expressed *Rpi-amr4-1102* in *N. benthamiana* leaves and inoculated the leaves with *P. infestans* zoospores. *Rpi-amr3* was used as a positive control and non-functional *Rpi-amr3a* from SP1102 was used as a negative control (Fig. 4f). This assay showed that *Rpi-amr4-1102* confers resistance against *P. infestans* isolate T30-4. However, the resistance was weaker than that with *Rpi-amr3* (Fig. 4f). We also generated stable *Rpi-amr4-1102* transgenic *N. benthamiana* lines. As expected, the T0 transgenic plants gained the capacity for PITG_22825 recognition, and were resistant to two *P. infestans* isolates T30-4 and 88069. We also verified this finding in the T1 *Rpi-amr4* transgenic lines (Supplementary Fig. 16a,b).

In summary, we successfully cloned a new *Rpi* gene *Rpi-amr4* from *S. americanum* and defined its cognate effector gene *Avramr4*

(*PITG_22825*). *Rpi-amr4* confers late blight resistance and may serve as a resource for producing late blight-resistant potatoes.

## Cloning of *R02860* and *R04373*

Although *Rpi-amr4* could be identified using a GWAS approach, the number of effectors recognized by a few *S. americanum* accessions was small and did not enable a clear GWAS signal. We therefore deployed BSA-RenSeq to clone two more immune receptors.

PITG_02860 (Fig. 5a) targets the host protein NRL1 and attenuates plant immunity and increases pathogen virulence, but the cognate receptor was unknown[40]. We found that PITG_02860 triggered HR in 5 of 52 *S. americanum* accessions, including SP2271. We tested an $F_2$ population of SP2271 (PITG_02860 responsive, R) × SP2272 (PITG_02860 non-responsive, NR), and found that recognition of PITG_02860 segregated according to a 3:1 ratio (104 R and 34 NR; $\chi^2$ (1, $N$ = 138) = 0.00966, $P$ = 0.92169) (Fig. 5b). The RenSeq pipeline was performed on the $F_2$ population, and most filtered SNPs were located within a 1-Mb region on the top of chromosome 4 of SP2271 (Fig. 5c). SCAR markers were designed based on the resequencing data and used for genotyping. The candidate gene was mapped to a 295 kb region between markers S42 and S36. Seven *NLRs* genes reside within the mapping interval and all belong to the *R3* family (Figs. 2a,b and 5c). To test these candidate genes, the ORFs from four candidate genes (*nlr13*, *nlr14*, *nlr16*, *nlr17*) were cloned into a binary vector under the control of the *35S* promoter and *Ocs* terminator and transformed into *Agrobacterium* for transient expression. The candidate genes were expressed alone or with PITG_02860 or AVRamr3 in *N. benthamiana* and *Nicotiana tabacum*. NLR16 and NLR17 were auto-active in *N. benthamiana*, but we found that co-expression of NLR16 and PITG_02860 activated HR in *N. tabacum* (Fig. 5d). To verify this finding, we generated *nlr16* knockout SP2271 lines by using the CRISPR–Cas9 system. As expected, the knockout lines lost recognition of PITG_02860 (Supplementary Fig. 17). Therefore, we conclude that NLR16 (R02860 hereafter) is the immune receptor for PITG_02860.

PITG_04373 (Fig. 5e) triggered HR in only 3 of 50 *S. americanum* accessions including in SP2300, which carries both functional *Rpi-amr1* and *Rpi-amr3* (Fig. 3). To clone the corresponding immune receptor gene, we first phenotyped a BC_1 backcross population of SP2271 (NR) × SP2300 (R). The BC1 population segregated for PITG_04373 responsiveness with a 1:1 ratio (198 R and 182 NR; $\chi^2$ (1, $N$ = 380) = 0.67368, $P$ = 0.41177) (Fig. 5f). The DNA from responsive or non-responsive progenies was bulked for BSA-RenSeq and most linked SNPs mapped to SP2271 chromosome 10 (Fig. 5g). SCAR markers were designed and an $F_2$ population of SP2271×SP2300 was phenotyped and genotyped. The PITG_04373 responsiveness was mapped to a 1.447-Mb interval with eight genes based on the SP2271 genome (Fig. 5g). Most candidates belong to the *Rpi-chc1* family, except an *R1* homolog (Fig. 5g). In the absence of a reference genome for SP2300, we used the SMRT-RenSeq assembly as the reference NLRome[29]. The SMRT-RenSeq contigs mapped to this region of the SP2271 genome, and candidate genes from SP2300 were cloned into a vector with the *35S* promoter and *Ocs* terminator for transient assays. Five candidate genes were tested (*C18.1*, *C18.2*, *C127*, *C168* and *C829*). We found that the candidate immune receptor C168 (R04373 hereafter) can specifically recognize PITG_04373 after transient expression in *N. benthamiana* (Fig. 5h). Therefore, we conclude R04373 is the immune receptor of PITG_04373.

SP2300 also carries functional *Rpi-amr1* and *Rpi-amr3* homologs. To test the function of R04373 in SP2300, we generated *Rpi-amr1-2300*/*Rpi-amr3-2300*/*R04373* triple knockout lines (Supplementary Fig. 18a). Forty transgenic SP2300 knockout lines were generated and phenotyped and 13 of these 40 knockout lines lost recognition of the three effectors (PpAVRamr1, AVRamr3 and PITG_04373). Two of these lines, SLJ25603#3 and SLJ25603#17, were genotyped and the knockout events were confirmed (Supplementary Fig. 18b–e). We also

co-expressed *R04373* with *PITG_04373*; however, the HR phenotype was not restored in these knockout lines (Supplementary Fig. 18e) and we hypothesized that the truncated *R04373* fragment might produce interfering RNAs. Interestingly, the triple knockout lines showed slightly elevated susceptibility to *P. infestans* (Supplementary Fig. 18f, g) compared to wild-type SP2300, but were more resistant than SP2271, suggesting that there are additional *Rpi* genes in SP2300.

To test the late blight resistance conferred by R02860 and R04373, we transiently expressed R02860, R04373, Rpi-amr4 and their combinations in *N. benthamiana* and measured *P. infestans* growth. However, although we observed a slight significant decrease in lesion size after transient expression of R02860 and R04373, the pathogen could still infect the plants. We also co-expressed Rpi-amr4 with R02860 or R04373 without enhancing the resistance of Rpi-amr4 (Supplementary Fig. 19). These results indicate that although R02860 and R04373 can recognize the RXLR effectors PITG_02860 and PITG_04373 from *P. infestans*, the resistance they confer can be overcome by *P. infestans*.

In summary, by using BSA-RenSeq, SMRT-RenSeq and map-based cloning strategies, we successfully cloned two new immune receptors, R02860 and R04373, and defined their recognized RXLR effectors, PITG_02860 and PITG_04373.

## Discussion

*Solanum* is the largest genus of the Solanaceae family, comprising more than 1,500 species, including many important crop plants such as potato, tomato and eggplant for which extensive genome sequence data are available. The *S. nigrum* complex is composed of many species with different ploidy levels, including *S. nigrum* (6×), *Solanum scabrum* (6×), *Solanum villosum* (4×) and *S. americanum* (2×). Some are regarded as weeds, but others are consumed as food and medicine in various countries[41]. Importantly, these species carry valuable genetic variation for resistance to diseases, including, but not limited to, potato late blight and bacterial wilt[25,29,31]. In this study, we sequenced and assembled four *S. americanum* genomes, and generated multi-omics datasets. These data enabled us to build an *S. americanum* pan-NLRome to study the evolution and function of the *NLR* genes in *S. americanum*.

Potato late blight triggered the Irish famine in the 1840s and remains a global challenge that greatly constrains potato production. To understand ETI of *S. americanum* to *P. infestans*, we used 'effectoromics' to dissect the ETI interactions between them. We generated a matrix of 315 RXLR effectors × 52 *S. americanum* accessions. Interestingly, AVRamr1 (36/52) and AVRamr3 (43/52) recognition is widely distributed in *S. americanum* accessions, indicating that *Rpi-amr1* and *Rpi-amr3* play important roles in the late blight resistance of *S. americanum*. This finding is consistent with the conclusions of a pan-genome ETI study of the interaction between *Arabidopsis* and *Pseudomonas syringae*[42]. Some effectors induce cell death in all *S. americanum* accessions, such as effectors in the AVRblb2 family (Fig. 3). This observation is consistent with previous findings that AVRblb2 (PexRD39/PexRD40) induces cell death in all tested wild potato species, but not in *N. benthamiana*[43,44]; this non-specific cell death might be a result of the virulence activities of AVRblb2. Some resistant accessions lack AVRamr1 and AVRamr3 recognition and thus are valuable sources of novel *Rpi* genes.

Three new immune receptors Rpi-amr4, R02860 and R04373 were cloned in this study (Figs. 4 and 5). We showed that *Rpi-amr4* elevates *P. infestans* resistance in *N. benthamiana*. PITG_02860 was reported to promote host susceptibility by targeting the host protein NRL1 (ref.40), and the virulence functions and host targets of AVRamr4 and PITG_04373 remain to be discovered. *P. infestans* is a fast-evolving pathogen and may be able to overcome single *Rpi* genes in the field within a few years. Resistance based on the recognition of a single effector can be easily overcome by mutations or silencing, as shown for *Avrvnt1*

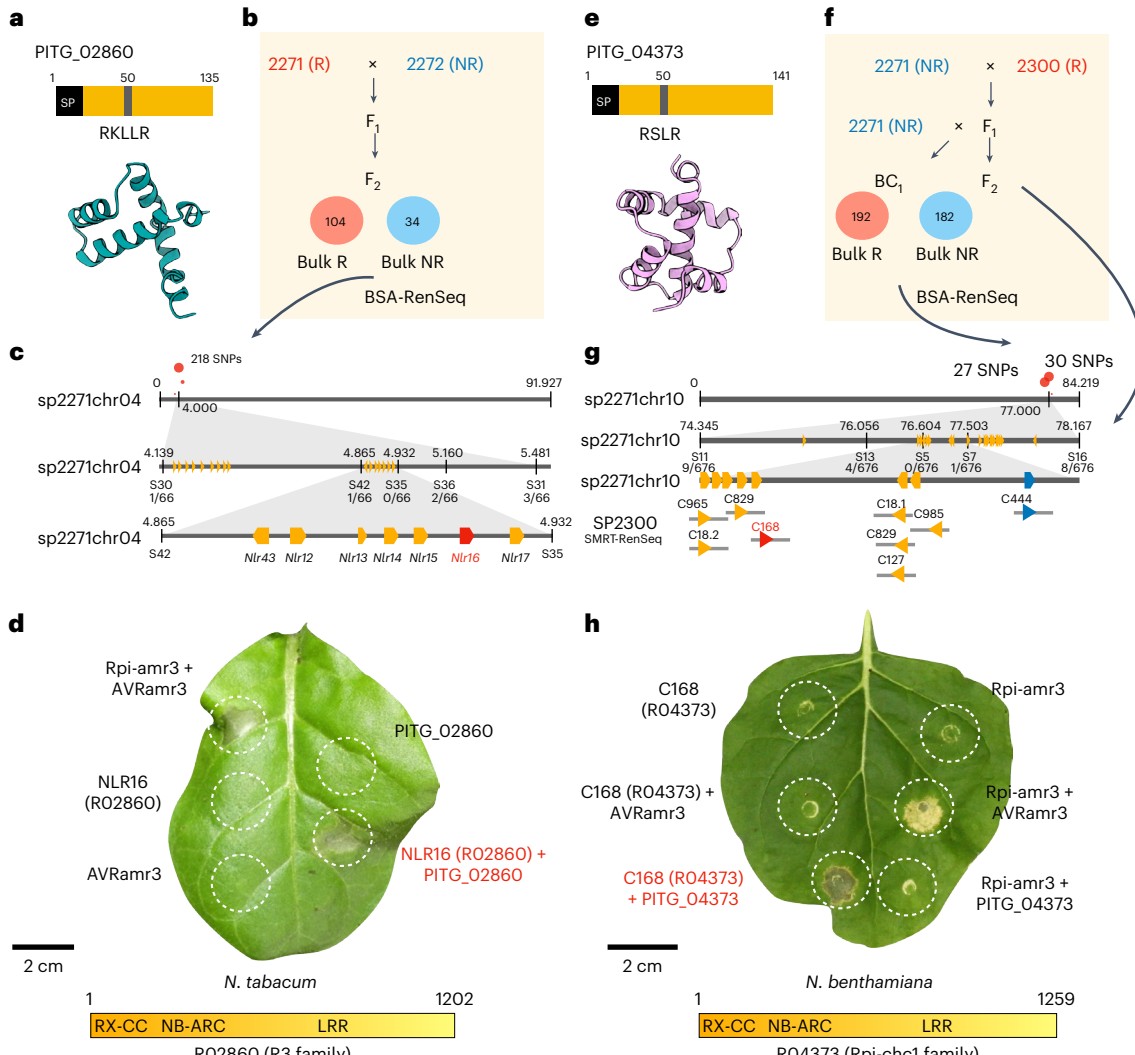

**Fig. 5 | Identification of R02860 and R04373 that recognize the RXLR effectors PITG_02860 and PITG_04373. a**, PITG_02860 is an RXLR effector from *P. infestans*. An illustration and predicted structure are shown. **b**, An F₂ population was generated from a cross between SP2271 (responds to PITG_02860, R) and SP2272 (no response to PITG_02860, NR). The R bulk (104 progenies) and NR bulk (34 progenies) were used for BSA-RenSeq. **c**, A total of 218 linked SNPs (red dots) on the top of chromosome 4 of SP2271 were identified. The gray bar represents the chromosome. The physical positions (in Mb) are shown by number. Five molecular markers (S30, S42, S35, S36 and S31) were used for the map-based cloning. The number of recombination events per total tested gametes is shown. **d**, HR assay of the candidate PITG_02860 receptor. The candidate genes were expressed alone, or co-expressed with *PVX::PITG_02860* in *N. tabacum* leaves. Rpi-amr3 and AVRamr3 were used as controls. NLR16 turned out to be the PITG_02860 receptor (R02860 hereafter). OD₆₀₀ = 0.5. Four-week-old *N. tabacum* plants were used, and the photos were taken at 4 dpi. Three biological replicates were performed with the same results.

**e**, PITG_04373 is an RXLR effector from *P. infestans*. An illustration and predicted structure are shown. **f**, Both backcross (BC₁) and F₂ populations were generated from SP2271 and SP2300. The BC₁ population of 192 responsive plants and 182 non-responsive progenies were bulked for BSA-RenSeq. The F₂ populations were used for fine mapping. **g**, Informative SNPs (red dots) on the top of chromosome 10 of SP2271 were identified. Five molecular markers (S11, S13, S5, S7 and S16) were used for fine mapping. The number of recombination gametes per total tested gametes is shown. Nine genes from SP2300 SMRT-RenSeq assemblies that mapped to the mapping interval of the SP2271 genome were regarded as candidate genes. All candidates belong to the *Rpi-chc1* family except *C444*. **h**, HR assay of the PITG_04373 receptor candidates. The candidate genes were expressed alone or co-expressed with *35S::PITG_04373* in *N. benthamiana* leaves, Rpi-amr3 and AVRamr3 were used as controls. C168 turned out to be the PITG_04373 receptor (R04373 hereafter). OD₆₀₀ = 0.5. Four-week-old *N. benthamiana* plants were used and photos were taken at 4 dpi. Three biological replicates were performed with the same results.

(refs.45,46). *R* gene stacking is a better way to deploy cloned *R* genes in the field[47,48]. Therefore, *Rpi-amr4* can be stacked with other *Rpi* genes to provide stronger and more durable potato late blight resistance.

The two other immune receptors R02860 and R04373, recognizing PITG_02860 and PITG_47373 were cloned from SP2271 and SP2300, respectively. The resistance conferred by these genes might be too weak to be applied in the field (Supplementary Fig. 19). Many effectors are suppressors of host immunity, notably AVRcap1, which can attenuate the function of the helper NLRs NRC2 and NRC3 (ref.49), and this attenuation may explain why some plant immune receptors

that recognize effectors nevertheless do not confer strong disease resistance. To understand the complex nature of plant–pathogen interaction, our work provides an assay to identify the suppressor of R02860 and R04373 in the future.

In summary, our study provides valuable genomic and genetics tools that should accelerate the path to understanding and achieving durable resistance against potato late blight and other plant diseases and shows that *S. americanum* is an excellent model plant to study molecular plant–microorganism interactions and plant immunity.

## Online content

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

## Methods

### Sequencing and assembly of *S. americanum* genomes

Four representative *S. americanum* accessions, SP1102, SP2271, SP2273 and SP2275, were selected for sequencing. The Pacific Bioscience Sequel II platform in the circular consensus sequencing (CCS) mode was applied to sequence the genomes of SP1102 and SP2271 and generated 30.5 Gb and 28.5 Gb of high-fidelity (HiFi) reads, respectively. The PromethION and GridION platforms of Oxford Nanopore Technologies were applied to sequence the genomes of SP2273 and SP2275, and generated ~81.1 Gb and ~114.5 Gb of data, respectively. To estimate the genome heterozygosity and polish the raw assembled genomes, we also prepared libraries for Illumina paired-end short-reads sequencing following the standard protocol and generated an average of 99.2 Gb of clean data for each *S. americanum* accession using the Illumina Hiseq 2500 platform. Hi-C libraries of three *S. americanum* accessions, SP1102, SP2271 and SP2273, were created from young seedlings based on the restriction enzyme MboI. The Illumina Hiseq 2500 platform was applied to generate 86.5, 81.8 and 54.8 Gb of paired-end reads for SP1102, SP2271 and SP2273, respectively.

The genome size and heterozygosity were estimated using a *k*-mer-based approach by KAT[50] and GenomeScope[51]. The estimated genome size was calculated as the total number of *k*-mers divided by the estimated sequencing coverage. The total number of k-mers could be calculated from sequencing data, and sequencing coverage could be assessed based on *k*-mer distribution frequency. In this study, we applied KAT to calculate *k*-mer frequency with $k = 19$ and the Perl script estimate_genome_size.pl (https://github.com/josephryan/estimate_genome_size.pl) to estimate the genome size of *S. americanum*. Hifiasm[52] was applied to assemble assemble SP1102 and SP2271 de novo using default parameters. To assemble the genomes of SP2273 and SP2275, we first corrected ONT reads using Canu[53] with parameters 'correct corOutCoverage=500 corMinCoverage=2 minReadLength=2000 genomeSize=1g -nanopore-raw'. The corrected reads were then assembled into raw contigs by SMARTdenovo[54] with the following command line arguments 'perl smartdenovo.pl -c 1 -t 24 -k 17'. The raw assemblies were then iteratively polished using Illumina short reads. Reads were aligned to the raw assemblies using BWA[55], and resulting bam files were passed to Pilon[56] for polishing. Pseudo-chromosomes were built by using the juicer[57] and 3d-DNA[58] pipeline with parameters '-m haploid -i 15000 -r 0'. The quality of the assemblies was assessed by BUSCO (Benchmarking Universal Single-Copy Orthologs)[59] with the solanales_odb10 database.

### Protein-coding genes prediction

For SP1102 and SP2271, to help with gene model prediction, the transcriptomes of *S. americanum* whole seedlings, roots, stems, leaves, flowers and fruits were sequenced by using the Illumina HiSeq 2500 platform with three replications for each tissue and 4 Gb of clean data for each sample. The reads were aligned to the genome by HISAT[60], transcripts were assembled using StringTie[61], Cufflinks[62] and Trinity[63] and the assemblies were then imported into PASA[64] for protein-coding gene prediction. Ab initio and homologous protein search strategies were also performed by using SNAP[65], AUGUSTUS[66], GlimmerHMM[67] and exonerate[68]. All predicted evidence was integrated using EVM[64]. To predict gene models in SP2273 and SP2275, we used the ITAG4.0 (ref.[14]) and SolTub_3.0 (ref.[13]) datasets for homology-based gene prediction in the GeMoMa program[69]. RNA-seq data, obtained from SP2273 were also incorporated for splice site prediction.

### Phylogenetic analysis of *S. americanum*

The representative protein sequences of *Arabidopsis thaliana*, *Petunia inflata*, *Capsicum annuum*, *Solanum melongena*, *Solanum tuberosum* Group Phureja (DM1-3 516 R44), *S. tuberosum* Group Tuberosum (RH89-039-16), *Solanum commersonii*, *Solanum chacoense*, *Solanum pennellii*, *Solanum pimpinellifolium*, *Solanum lycopersicum*

and four *S. americanum* accessions (SP1102, SP2271, SP2273 and SP2275) were extracted and input into OrthoFinder[70] to cluster orthogroups using the MCL algorithm. The protein sequences of 1,363 single-copy orthogroups were extracted to infer the phylogenetic relationship following the supermatrix method. Sequences from 15 genomes were aligned using MAFFT[71] with parameter '--auto' and trimmed using trimAl with parameters '-phylip -gt 0.8'. IQ-TREE[72] was applied to infer the phylogenetic relationships with parameters '--alrt 1000 -B 1000'. We used BASEML and MCMCTREE from the PAML software package[73] to estimate the divergence time. The coding sequence (CDS) sequences of 1,363 single-copy orthogroups were extracted for a rough estimation of the substitution rate using BASEML with model = 7. MCMCTREE with parameters 'model = 7, burnin = 5,000,000, sampfreq = 300, nsample = 20,000' was applied to estimate the divergence time. The divergence times of potato–tomato (7–10 Ma)[34] and potato–*Arabidopsis* (111–131 Ma; http://www.timetree.org/) were used for calibration. Two rounds of estimation were performed with similar results.

### Genomic alignments of *S. americanum* and neighboring species

Genomic alignment between SP1102 and eggplant/potato was performed using MUMMER[74] with parameters '--maxmatch -c 100 -b 500 -l 50'. The alignment was further filtered with parameters '-1 -i 80 -l 100'. The delta format was then converted to PAF format using the paftools.js script[75] and passed to D-Genies[76] for dot plot visualization.

### Syntenic analysis of *S. americanum* and neighboring species

We applied the Python-based program MCscan (v1.1.8) (ref. [77]) to perform the syntenic analysis. The representative protein sequences and corresponding gene model annotations in BED format of potato (DMv6.1), eggplant (HQ-1315) and four *S. americanum* genomes were extracted to search for homologous with parameters '-m jcvi.compara.catalog ortholog --cscore = .99'. Syntenic regions were identified with '-m jcvi.compara.synteny screen --minspan=30' and visualized with '-m jcvi.graphics.karyotype' parameters.

### Identification of SVs

To identify large SVs (>1 Mb in length), we aligned the chromosome-grade assembly (SP2271 and SP2273) to the SP1102 reference genome by using MUMMER with parameters: '--batch 1 -t 20 -l 100 -c 500' and further filtered the alignment with parameters: '-i 90 -l 100'. SyRI v1.4 was adopted to identify SVs based on the alignment delta files; only large SVs were kept for further analysis. We adopted the Hi-C interaction map and SV location to validate the large SVs. Of the 70 SVs identified in SP2271 and SP2273, 68 SVs reside in single contig, suggesting high reliability. Of these, 40 SVs could be verified by a Hi-C interaction map.

The assembly-based approach was applied to identify SVs (> 40 bp in length) among *S. americanum* genomes following the pipeline of SVIM-asm[78]. The contig assemblies of SP2271, SP2273 and SP2275 were aligned to the SP1102 reference using minimap2 (ref.[76]) with the following parameters '--paf-no-hit -a -x asm5 --cs -r2k'. SVs, which consist of insertions, deletions, duplications and inversions were identified using SVIM-asm with 'haploid' mode. The SVs were further annotated by SnpEff[79].

### Calculation of sequence diversity across the *S. americanum* genome

We used the SV information generated by SVIM-asm and a sliding-window (window = 500 kb, step = 50 kb) method to calculate sequence diversity across the *S. americanum* genome. The diversity of a window was calculated as follows: diversity = (sum of SV length in a window) / window length. The final diversity value of each window was generated from the average of SP2271, SP2273 and SP2275. Higher diversity values refer to higher variation levels of a window. If a window

overlapped an *NLR* gene, this window was counted as an *NLR*-region and a total of 2,304 *NLR*-regions were extracted from the SP1102 genome. To compare the variations between *NLR*-region and non-*NLR*-regions, we randomly selected 2,304 non-*NLR*-regions and compared their diversity values with those for *NLR*-regions by Wilcoxon rank-sum test. Ten rounds of random selection and comparisons were done between non-*NRL*-regions and *NLR*-regions.

## Annotation of *NLR* gene models

The *NLR* genes from the four *S. americanum* reference genomes were predicted by NLR-annotator[80]. To obtain a better gene model for these *NLR* genes, all the *NLR* genes from SP2271, SP1102 and SP2273 genomes were manually curated. In brief, the outputs of NLR-annotator were imported into Geneious (v10.2.6) (ref. 81) as annotations of the reference genomes and the predicted *NLR* fragments with 2 kb flanking sequences from both sides were extracted. Augustus[66] was then used to predict the gene model based on the trained dataset of tomato. The gene models were curated based on functionally validated *NLR* genes from public databases and cDNA RenSeq data were also used to assist the manual annotation.

## Phylogenetic analysis of NLRs in *S. americanum* genomes

To infer the phylogeny of NLRs, the protein sequences for the NB-ARC domain found using NLR-annotator were aligned using MAFFT[71] and IQ-TREE was used to build a phylogenetic tree. The JTT + F + R9 substitution model was selected by ModelFinder[82] and used to infer the maximum-likelihood tree. Ultrafast bootstrap (UFBoot)[83] was set to 1,000. *CED-4* from *C. elegans* was selected as an outgroup.

To analyze the NLR presence and absence in *S. americanum* genomes, we collected 13 previously reported SMRT RenSeq assemblies[37] and generated 3 new assemblies from accessions SP2298, SP3370 and SP2308. NLR-annotator was used to annotate the *NLR* genes in the SMRT-RenSeq dataset. We used GMAP[84] to predict the NLR homologs among the *S. americanum* genomes and SMRT-RenSeq assemblies. The CDS sequences of manually curated NLRs in the SP1102 genome were extracted and mapped to the three *S. americanum* genomes and 16 SMRT RenSeq assemblies using GMAP with parameters '-f 2 -n 1 --min-trimmed-coverage=0.70 --min-identity=0.70'. NLRs that failed to align were marked as absent. In the v4 RenSeq library[85], more baits were included compared to the v3 RenSeq library; thus, if a certain NLR was absent in all v3 RenSeq assemblies but present in v4 assemblies, the absence might be a false-positive and was marked as NA. To calculate the expression level of *NLR* genes in SP1102, RNA was isolated from young leaves of SP1102 and cDNA RenSeq was done as described previously[86]. We mapped the reads from SP1102 cDNA RenSeq to its genome using STAR (2.6.0c) (ref. 87), the BAM files were imported into Geneious (v10.2.6) (ref. 81) and the TPM values for *NLR* genes were calculated using the 'Calculate Expression Levels' function. The NLR phylogeny, TPM and PAV results were passed to the online software iTOL[88] for final visualization.

## Analysis of the *S. americanum* pan-NLRome

The NLR protein sequences from 4 genome assemblies as well as 16 SMRT-RenSeq assemblies were classified into orthogroups by OrthoFinder using the MCL algorithm. The orthogroups matrix was then processed with PanGP (v.1.0.1) (ref. 89) using the random algorithm. The sample size and sample repeat parameters were set to 500 and 30, respectively. These parameters indicate that at each given accession number (*n*), *n* accessions will be randomly selected for pan- and core-NLR analysis. A 500 times random selection was performed with 30 replicates. The estimated size of pan- and core-NLRomes were illustrated with a box plot and fitted with exponential models. The orthogroups were classified into three categories according to their frequency of occurrence: core (orthogroups present in all 18–20 accessions); dispensable (orthogroups that were missed in more than 3

accessions and present in at least 2 accessions); and unique (orthogroups present in only 1 accession). For each accession, the numbers of NLRs in different categories were summarized and illustrated with a stacked bar chart.

## Resequencing of 52 *S. americanum* accessions

The genomic DNA of 4-week-old young leaves from 52 *S. americanum* accessions was sampled and isolated using a Qiagen DNeasy plant kit (Qiagen, 69104). A whole-genome PCR-free, 2 × 150 bp paired-end Illumina library was generated and sequenced by Novogene (Beijing, China), generating ~10 Gb of data for each *S. americanum* accession. The raw reads were trimmed using trimmomatic v0.36 (ref. 90). The clean reads of each accession were mapped to the SP1102 reference genome with minimap2 (v2.16) (ref. 75), and converted to BAM format with samtools (v1.9). SNP calling was carried out with samtools and bcftools (v1.9).

To infer the phylogenetic relationships of *S. americanum* accessions, we selected the genomes of potato (DM 1-3 516 R44 v6.1), tomato (Heinz 1706 v4.0) and eggplant (v3) as an outgroup. Wgsim (https://github.com/lh3/wgsim) was used to simulate whole-genome sequencing reads from the potato, tomato and eggplant genomes with parameters: '-e 0 -d 350 -N 500000000 -1 150 -2 150 -r 0 -R 0 -X 0'. The simulated reads mapping and SNP calling were performed using the same approaches. Bedtools (v2.17) was used to extract SNPs in coding regions. The SNP-based phylogenetic tree was inferred by IQ-TREE with UFBoot set to 1,000 and the TVMe+R2 best-fit model, which was automatically selected by ModelFinder. The phylogenetic tree was visualized with FigTree (v1.4.4).

## GWAS analysis

For the GWAS analysis, all the SNPs residing in *NLR* gene regions, as well as the 3 kb upstream and 1 kb downstream regions, were extracted with bedtools (v2.17). The SNPs were filtered and processed using Plink (v1.90) with parameters '--make-bed --allow-extra-chr --allow-no-sex --mind 1 --maf 0.05 -geno 0.05 --recode --out'. The responsiveness scores of each effector were used as the phenotype and passed to Plink for association analysis with parameters '--allow-extra-chr --allow-no-sex --assoc --bfile --pheno'. The Manhattan plot was created using the R package qqman (v0.1.8).

## Effectoromics screening

An RXLR effector library of 311 RXLR effectors was used in the effectoromics screening. The signal peptides were removed, and the effector domains were cloned into overexpression vectors (pMDC32 or pICSL86977) or PVX vectors. *S. americanum* plants were grown in a containment glasshouse. Four- or 5-week-old plants were used for the agroinfiltration. For the overexpression vectors, cell death was scored at 4 dpi; for the PVX vectors, cell death was scored at 7 dpi. $OD_{600}$ = 0.5. The cell death phenotype was scored (2, strong HR; 1, weak HR; 0, no HR). Two leaves each from two plants were used for each experiment.

## Constructs for transient overexpression

To verify the candidate genes, the ORF of each candidate genes was amplified by Phusion high-fidelity DNA polymerase (NEB, M0530S) or KAPA HiFi Uracil+ DNA polymerase (Roche, 07959052001) and then cloned into the pICLS86922 overexpression vector with the *35S* promoter and *Ocs* terminator using BsaI (NEB, #R3733) or a USER cloning vector (pICSLUS0004OD) with the *35S* promoter and *Ocs* terminator using USER enzyme (NEB, #M5508). The verified constructs were transformed into *Agrobacterium* for transient expression in plants.

## Gene knockout with the CRISPR–Cas9 system

For the knockout constructs, guide RNAs were designed in Geneious (v10.2.6) using the 'Find CRISPR Site' function with parameters: 'Maximum mismatches allowed against off-targets = 3; Maximum

mismatches allowed to be indels = 0; pair CRISPR Sites: Maximum overlap of paired sites = 100; Maximum allowed space between paired sites = 300'. The reference genome or SMRT-RenSeq assembly was used for scoring of off-target activity. The selected guide RNAs are shown in Supplementary Table 3. Two guide RNAs for each candidate gene were amplified with the sgRNA scaffold by Q5 high-fidelity DNA polymerase (NEB, M0491S) and the pICSL70001 vector was used as the template. The fragments were then fused with an *Arabidopsis U6-26* promoter (pICL90002) and cloned into level 1 vectors at different positions (position 3, pICH47751; position 4, pICH47761; position 5, pICH47772; position 6, pICH47772). For the final level 2 constructs, *Cas9* with introns (position 1, pICSL11197), NPTII (position 2, pICSL11055), an end linker (pICH41922) and the guide RNAs were assembled into pICSL4723_OD. The final constructs were then transformed into *S. americanum* accessions from gene knockout. After transformation, the T$_0$ lines were moved into a containment glasshouse for phenotyping and genotyping. Agroinfiltration of the corresponding effector was used for the phenotyping. Genomic DNA from the individual T0 lines was isolated, and specific primers were designed for the *Cas9* gene and the target genes. Amplicons from the target genes were sub-cloned into the pGEM-Teasy TA cloning vector (Promega, A1360) or pICSL86977 for sequencing. The sequencing data were analyzed in Geneious (v10.2.6).

### Plant growth and transformation

The *N. benthamiana* and *N. tabacum* cv. Petit Gerard plants were sowed and grown in a controlled environment room (CER) at 22 °C and 45–65% humidity with a 16-h photoperiod. Four-week-old plants were used for the HR assay.

For the *S. americanum* transformation, sterilized seeds (SP2271 and SP2300) were sown in MS medium (2% sucrose). Leaf disks were cut from 4 to 6-week-old in-vitro plants. Overnight *Agrobacterium* (AGL1) culture (100 µl) and 200 µM acetosyringone were added to 20 ml of LSR broth and the leaf discs were gently dipped into the solution using sterile forceps for 20 min. The leaf discs were then removed from the *Agrobacterium* suspension, blotted dry, and incubated under low light conditions at 18–24 °C for 3 days. The dried leaf discs were plated on LSR1 + 200 µM Acetosyringone solid medium. Co-cultivated explants were transferred to LSR1 medium in petri dishes with selection antibiotics (about seven leaf discs per plate). Explants were subcultured onto fresh LSR1 medium approximately every 14 days. Once the calli were sufficiently developed they were transferred onto LSR2 medium. Subculturing continued every 14 days when shoots started to appear. The shoots were removed with a sharp scalpel and planted into MS2R solid medium with selection antibiotics. Transgenic plants harboring appropriate antibiotic or herbicide resistance genes should root normally by the fourth week and can then be weaned out of tissue culture into sterile peat blocks before being transferred to the glasshouse. Media used had the following components: LSR broth (1× MS medium, 3% sucrose, pH 5.7); LSR1 medium (1× MS medium, 3% sucrose, 2.0 mg L$^{-1}$ zeatin riboside, 0.2 mg L$^{-1}$ NAA, 0.02 mg L$^{-1}$ GA3, 0.6% agarose, pH 5.7); LSR2 medium (1× MS medium, 3% sucrose, 2.0 mg L$^{-1}$ zeatin riboside, 0.02 mg L$^{-1}$ GA3, 0.6% agarose, pH 5.7); MS2R (1× MS medium, 2% Sucrose, 100 mg L$^{-1}$ myo-inositol, 2.0 mg L$^{-1}$ glycine, 0.2% Gelrite, pH 5.7).

### Disease assay

*P. infestans* isolates T30-4 and 88069 were used for the disease test and were maintained on rye sucrose agar (RSA) medium in an 18 °C incubator. To induce zoospores, ice-cold water was added to the plates after 10–14 days. The plates were then incubated at 4 °C for 1–2 h and a hemocytometer was used to count the number of zoospores. The zoospore suspension was used for the DLA (100–500 zoospores per droplet).

### BSA-RenSeq and map-based cloning

Three mapping populations were used in this study: (1) F$_2$ populations of SP2271 × SP2272 and (2) BC$_1$ and (3) F$_2$ populations of SP2271 × SP2300. The populations were phenotyped by agroinfiltration of RXLR effectors. A cork borer was used for sampling, and the leaf discs from the responsive and non-responsive progenies were pooled. The Genomic DNA was isolated using the Qiagen DNeasy plant kit (Qiagen, 69104). RenSeq libraries were then prepared, as described previously[24]. The libraries were sequenced (Illumina 2 × 250-bp reads) by Novogene (Beijing, China). The SNP filtering and calling steps were described previously[26].

To design molecular markers, the 10× PCR-free resequencing reads were mapped to the SP2271 *S. americanum* reference genome. Then SCAR markers that linked with the BSA-RenSeq signals were designed; amplicons should only be present in the non-responsive allele. The SCAR markers were first tested on the parental lines and the verified markers were then used on genomic DNA from individual non-responsive plants. GoTaq G2 DNA polymerase (Promega, 0000066542) was used for genotyping.

### Reporting summary

Further information on research design is available in the Nature Portfolio Reporting Summary linked to this article.

## Data availability

The raw sequencing data and genome assemblies of SP1102, SP2271, SP2273 and SP2275 genomes have been deposited at the National Center for Biotechnology Information (NCBI) Sequence Read Archive (SRA) with BioProject accession number PRJNA845062; The raw SMRT RenSeq data were deposited in ENA under project number PRJEB38240. The whole-genome resequencing data were deposited in ENA under project number PRJEB57057. The BSA-RenSeq data were deposited in ENA under project numbers PRJEB57070 and PRJEB57074. The assembled genomes, gene structure annotations, SMRT-RenSeq assemblies and manually annotated *NLR* genes as well as variation information are available at Figshare (https://figshare.com/projects/The_Solanum_americanum_pangenome_and_effectoromics_reveals_new_resistance_genes_against_potato_late_blight/145449). The *SaNRC1-1102*, *SaNRC2-1102*, *SaNRC3-1102*, *Rpi-amr4-1102*, *Rpi-amr4-2271*, *RO2860* (*Rpi-amr16*) and *RO4373* (*Rpi-amr17*) sequences were deposited at NCBI GenBank under accession number OP918030–OP918036. Source data are provided with this paper.

## Code availability

Custom scripts and codes used in this study are available at Zenodo (https://doi.org/10.5281/zenodo.7928678)[91].

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

## Acknowledgements

This research was financed by BBSRC grants BB/P021646/1 (J.D.G.J.), BB/S018832/1 (J.D.G.J.), BB/W017423/1 (J.D.G.J.), the Gatsby Charitable Foundation (J.D.G.J.), National Key Research and Development Program of China (2019YFA0906200, S.H.), Guangdong Major Project of Basic and Applied Basic Research (2021B0301030004, S.H.), Agricultural Science and Technology Innovation Program (CAAS-ZDRW202101, S.H.), Shenzhen Outstanding Talents Training Fund (S.H.), National Research Foundation of Korea (2018R1A5A1023599 and 2023R1A2C3002366, K.H.S.) and New Breeding Technologies Development Program (PJ015799 and PJ016538, S.J.P.). We thank TSL transformation team (A. Wawryk-Khamdavong), SynBio team (M. Youles and L. Egan), media services (N. Stammars), bioinformatics team (D. MacLean and C. Jégousse) and horticultural team (S. Perkins, J. Smith, L. Phillips, C. Taylor, T. Wells, D. Alger and S. Able) for their support. We thank P. Robinson (JIC) for the scientific photography. We thank S. Marillonnet (Icon Genetics GmbH, Halle/Saale, Germany) for sharing the Cas9 construct (pAGM47523). We thank Experimental Garden and Genebank of Radboud University, Nijmegen, the Netherlands, IPK Gatersleben, Germany and S. Knapp (Natural History Museum, London, UK) for access to *S. americanum* genetic diversity. We thank V. G. A. A. Vleeshouwers at Wageningen University and Research, P. Birch, I. Hein and B. Harrower at James Hutton Institute for making available clones of some effectors. We thank B. B.H. Wulff, S. Arora and K. Gaurav (JIC) for helpful discussions.

## Author contributions

X.L. Y.J., K.H.S., S.H. and J.D.G.J. conceived and designed the project. X.L. and Y.J. wrote the first draft with input from all the authors. X.L., Y.J., K.H.S., S.H. and J.D.G.J. reviewed and edited the manuscript. Y.J., M.P., X.L., R.K.S. and J.H. performed the bioinformatics analyses. X.L. and M.M. performed the effectoromics screening. X.L., R.H., M.S., A.C.O.A., S.F. and A.N. contributed to cloning and characterization of *Rpi-amr4*, *RO2860* and *RO4373*. M. Smoker and J.T. performed the plant transformation. K.W., Y.L., C.Z., S.J.P., K.H.S., S.H. and J.D.G.J. contributed resources.

## Competing interests

The authors declare no competing interests.

## Additional information

**Correspondence and requests for materials** should be addressed to Xiao Lin, Kee Hoon Sohn, Sanwen Huang or Jonathan D. G. Jones.

# Reporting Summary

## Statistics

For all statistical analyses, confirm that the following items are present in the figure legend, table legend, main text, or Methods section.

| n/a | Confirmed | |
|---|---|---|
| ☐ | ☒ | The exact sample size (*n*) for each experimental group/condition, given as a discrete number and unit of measurement |
| ☐ | ☒ | A statement on whether measurements were taken from distinct samples or whether the same sample was measured repeatedly |
| ☐ | ☒ | The statistical test(s) used AND whether they are one- or two-sided *Only common tests should be described solely by name; describe more complex techniques in the Methods section.* |
| ☒ | ☐ | A description of all covariates tested |
| ☐ | ☒ | A description of any assumptions or corrections, such as tests of normality and adjustment for multiple comparisons |
| ☐ | ☒ | A full description of the statistical parameters including central tendency (e.g. means) or other basic estimates (e.g. regression coefficient) AND variation (e.g. standard deviation) or associated estimates of uncertainty (e.g. confidence intervals) |
| ☐ | ☒ | For null hypothesis testing, the test statistic (e.g. *F*, *t*, *r*) with confidence intervals, effect sizes, degrees of freedom and *P* value noted *Give P values as exact values whenever suitable.* |
| ☒ | ☐ | For Bayesian analysis, information on the choice of priors and Markov chain Monte Carlo settings |
| ☒ | ☐ | For hierarchical and complex designs, identification of the appropriate level for tests and full reporting of outcomes |
| ☒ | ☐ | Estimates of effect sizes (e.g. Cohen's *d*, Pearson's *r*), indicating how they were calculated |

*Our web collection on statistics for biologists contains articles on many of the points above.*

## Software and code

Policy information about availability of computer code

| | |
|---|---|
| Data collection | No software was used for data collection. |
| Data analysis | KAT (v2.4.2), GenomeScope (v2.0), Hifiasm (v.0.13), juicer (v.1.5), 3d-DNA (v.180922), Canu (v1.8), SMARTdenovo (v8488de9), BWA (0.7.5a-r405), Pilon (v.1.23), BUSCO (), HISAT (v.2.0.1-beta), StringTie (v.1.3.3b), Cufflinks (v2.2.1), Trinity (v2.10.0), PASA (v2.4.1), SNAP (v.2013-02-16), AUGUSTUS (v.3.4.0), GlimmerHMM (v3.0.4), exonerate (v2.2), EVM (v1.1.1), GeMoMa (v1.7.1), OrthoFinder (v2.5.2), MAFFT (v.7.471), trimAl (v1.4.1), IQ-TREE (v2.1.4-beta), BASEML (v.4.9), MCMCTREE (v.4.9), MUMMER (v4.0.0rc1), D-Genies (v1.2.0), MCscan (Python version), Python (v3.5.6), minimap2 (v2.17-r941), SVIM-asm (v1.0.2), SnpEff (v5.0e), Geneious (v10.2.6), GMAP (v2020-10-14), STAR (2.6.0c), iTOL (v5), trimmomatic (v0.36), samtools (v1.9), bcftools (v1.9), Wgsim (2011 version), Bedtools (v2.17) , FigTree (v1.4.4), Plink (v1.90), R package qqman (v0.1.8), clinker (2020 version). <br><br> Some customized Python scripts were used to process the data generated by each software, which parameters were described in Methods section. All codes are available from the corresponding author upon request. |

For manuscripts utilizing custom algorithms or software that are central to the research but not yet described in published literature, software must be made available to editors and reviewers. We strongly encourage code deposition in a community repository (e.g. GitHub). See the Nature Portfolio guidelines for submitting code & software for further information.

## Data

Policy information about availability of data

All manuscripts must include a data availability statement. This statement should provide the following information, where applicable:
- Accession codes, unique identifiers, or web links for publicly available datasets
- A description of any restrictions on data availability
- For clinical datasets or third party data, please ensure that the statement adheres to our policy

The raw sequencing data for SP1102, SP2271, SP2273 and SP2275 genomes have been deposited at the National Center for Biotechnology Information (NCBI) Sequence Read Archive (SRA) with BioProject accession number PRJNA845062 (https://dataview.ncbi.nlm.nih.gov/object/PRJNA845062? reviewer=hliiufd2hm679172evsbdgcr69); The raw SMRT RenSeq data were deposit in ENA under project number: PRJEB38240; The whole genome resequencing data were deposit in ENA under project number: PRJEB57057; The BSA-RenSeq data were deposit in ENA under project number: PRJEB57070 and PRJEB57074. The assembled genomes, gene structure annotations, SMRT RenSeq assemblies, manually annotated NLR genes as well as variation information are available at Figshare (https://figshare.com/projects/The_Solanum_americanum_pangenome_and_effectoromics_reveals_new_resistance_genes_against_potato_late_blight/145449). The SaNRC1-1102, SaNRC2-1102, SaNRC3-1102, Rpi-amr4-1102, Rpi-amr4-2271, R02860 (Rpi-amr16) and R04373 (Rpi-amr17) sequences were deposited at NCBI GenBank under accession number: OP918030-OP918036.

## Human research participants

Policy information about studies involving human research participants and Sex and Gender in Research.

| | |
|---|---|
| Reporting on sex and gender | N/A |
| Population characteristics | N/A |
| Recruitment | N/A |
| Ethics oversight | N/A |

Note that full information on the approval of the study protocol must also be provided in the manuscript.

# Field-specific reporting

Please select the one below that is the best fit for your research. If you are not sure, read the appropriate sections before making your selection.

☒ Life sciences　　　☐ Behavioural & social sciences　　　☐ Ecological, evolutionary & environmental sciences

For a reference copy of the document with all sections, see nature.com/documents/nr-reporting-summary-flat.pdf

# Life sciences study design

All studies must disclose on these points even when the disclosure is negative.

| | |
|---|---|
| Sample size | No statistical analysis was used to determine the sample size. The sample size is widely used and accepted in the field, the details were described in the Methods section. |
| Data exclusions | For the disease assay in Figure 4d, the preliminary disease assay was not blinded and with a small sample size. The results were consistent with the 4 blind replicates, but the preliminary result was not included in the final statistical analysis and figure. |
| Replication | For Fig. 3: 3 biological replicates were performed for all the responsive effectors.<br>For the HR assays in Fig. 4 and Fig. 5: 3 biological replicates were performed.<br>For the disease assay in Figure 4d: 4 biological replicates were performed.<br>For the disease assay in Figure S19: 4 biological replicates were performed. |
| Randomization | All the plants in the same experiment were grew in the same condition. All samplings were randomized. |
| Blinding | For the disease assay in Figure 4d: Blind tests were performed for all the 4 replicates. For each replicate, a colleague streaked out the constructs (Rpi-amr3, Rpi-amr3a and Rpi-amr4) with a random code "A", "B" or "C".  Another researcher performed the agroinfiltration, detached leaf assays, and scoring. The genotype codes were revealed after scoring. |

# Reporting for specific materials, systems and methods

We require information from authors about some types of materials, experimental systems and methods used in many studies. Here, indicate whether each material, system or method listed is relevant to your study. If you are not sure if a list item applies to your research, read the appropriate section before selecting a response.

## Materials & experimental systems

| n/a | Involved in the study |
|-----|------------------------|
| ☒ | ☐ Antibodies |
| ☒ | ☐ Eukaryotic cell lines |
| ☒ | ☐ Palaeontology and archaeology |
| ☒ | ☐ Animals and other organisms |
| ☒ | ☐ Clinical data |
| ☒ | ☐ Dual use research of concern |

## Methods

| n/a | Involved in the study |
|-----|------------------------|
| ☒ | ☐ ChIP-seq |
| ☒ | ☐ Flow cytometry |
| ☒ | ☐ MRI-based neuroimaging |

