## [Peer Review File · Nature Genetics]

Peer Review Information

Manuscript Title: Solanum americanum genome-assisted discovery of immune receptors that detect potato late blight pathogen effectors

Corresponding author name(s): Professor Jonathan Jones, Dr Xiao Lin, Professor Sanwen Huang, Kee-Hoon Sohn

Reviewer Comments & Decisions:

Decision Letter, initial version:

6th Oct 2022

Dear Professor Jones,

Your Article, "The Solanum americanum pangenome and effectoromics reveal new resistance genes against potato late blight" has now been seen by 4 referees. You will see from their comments below that while they find your work of interest, some important points are raised. We are interested in the possibility of publishing your study in Nature Genetics, but would like to consider your response to these concerns in the form of a revised manuscript before we make a final decision on publication.

To guide the scope of the revisions, the editors discuss the referee reports in detail within the team with a view to identifying key priorities that should be addressed in revision. In this case, we think all referees have identified important aspects of the genomic analyses and the experiments that need to be improved or clarified. We particularly ask that you strengthen the pan-genome analysis, provide more functional evidence to support the role of the newly identified resistance genes, and address all referee comments as thoroughly as possible with appropriate revisions. We hope that you will find the prioritized set of referee points to be useful when revising your study.

We therefore invite you to revise your manuscript taking into account all reviewer and editor comments. Please highlight all changes in the manuscript text file. At this stage we will need you to upload a copy of the manuscript in MS Word .docx or similar editable format.

*2) If you have not done so already please begin to revise your manuscript so that it conforms to our Article format instructions, available [here](http://www.nature.com/ng/authors/article_types/index.html). Refer also to any guidelines provided in this letter.

[redacted]

We hope to receive your revised manuscript within 3 to 6 months. If you cannot send it within this time, please let us know.

Sincerely,
Wei

Wei Li, PhD

Senior Editor
Nature Genetics
New York, NY 10004, USA
www.nature.com/ng

Reviewers' Comments:

Reviewer #1:

Remarks to the Author:

Lin et al. described the genetic variation and characterization of NLR immune receptor genes of *Solanum americanum* by constructing a pan-genome of *S. americanum*, and performing an effectoromics screening against 315 *Phytophthora infestans* RXLR effectors in 52 *S. americanum* accessions. Based on resistant phenotype of *S. americanum* accessions against potato late blight, the authors selected and assembled four high-quality de novo genome assemblies. Re-sequencing of 52 *S. americanum* accessions with SMRT-RenSeq revealed phylogenetic relationship, population structure, and the genetic variation of NLRs. Screening of 315 *P. infestans* RXLR effectors in 52 *S. americanum* accessions elucidated the effector-triggered immunity landscape between *S. americanum* and *P. infestans*. Using GWAS, BSA-RenSeq, SMRT-RenSeq, and map-based cloning strategies, three novel Rpi-Avr gene pairs were characterized. Overall, these results and resources are comprehensive and should be very informative to the community. However, there are some major and minor concerns that should be resolved before consider publication of the manuscript.

Major concerns

1. Although the title includes the word "pangenome", these genomic information does not cover whole genetic variation of *S. americanum* species. The phylogeny and population structure of 52 *S. americanum* accessions separated into four groups (Supplementary Fig. 12a). However, the authors selected four accessions in group 1 to 3 to construct pan-genome. These accessions do not cover the genetic variation of its own group and group 4 (for example, population structure of SP2298 to SP2302 in group 3 and all accessions in group 4). If the "pangenome" mean "pan-NLRome", the title should be changed. If "pan-NLRome" is what the authors intended to, the saturation curves for core- and pan-NLR numbers to confirm that the pan-NLRome covers the entire NLR repertoire of *S. americanum* should be addressed.
2. The authors selected four *S. americanum* accessions based on phenotypic variation in resistance to late blight (Line 75 and Supplementary Fig. 1) but the Supplementary Fig. 1 do not show resistance phenotype. I recommend to include in detail description and phenotype images in the figure.
3. As the authors described (line 197-198), some accessions of *S. americanum* are susceptible to *P. infestans* (I couldn't understand meanings of the sentence "yet resistant to late blight in the field", Is it by inoculation? or in the natural condition?). Moreover, Witek et al. (2021, Nature Plants) also described the accession SP2271 as a susceptible host of *P. infestans* but the authors described *S. americanum*-*P. infestans* interaction as a nonhost pathosystem (in Line 13, 61, and 433). However, nonhost resistance is often defined as "a resistance of an entire species of plant against an entire

species of pathogen". Thus, *S. americanum* in this study could not fit the case of "nonhost resistance" against *P. infestans*. This should be considered and resented in the text.

4. Among the newly identified three NLRs (Rpi-amr4/16/17), the authors showed only the case of Rpi-amr4 exhibited partial/quantitative resistance against *P. infestans* and clearly confirmed the relationship between resistance phenotype of *S. americanum* against *P. infestans* and Rpi-amr4 using knockout lines and complementation assay. However, for the cases of Rpi-amr16 and 17, the authors presented each NLRs could induce HR-like cell death against corresponding effectors without any evidences on the blight resistance in planta. Quite a few of those cases are not end-up with actual disease resistance in plants. Thus, at least, transient expression-mediated resistance against *P. infestans* should be presented using Rpi-amr16/17 (because of the autoactive nature of those NLR with strong promoter such as 35S, authors may need its native promoter).

5. Data availability: There are missing datasets. The manual annotation of NLRs was performed to SP1102, SP2271, and SP2273, but only SP1102 was provided (Supplementary Dataset 1 to 3). Resequencing data of 52 *S. americanum* accessions and SMRT RenSeq data of SP2298, SP3370, and SP2308 also missed. Furthermore, I strongly recommend uploading of the variant calling file (VCF) to Figshare.

Minor Concerns

1. Supplementary Fig. 6: It would be better to state or describe that the reference is SP1102.
2. Line 145-147: The paragraph described the diversity of NLR region compared to non-NLR region. This is thought to be more related to the host resistance by NLR diversity than to the "nonhost" resistance.
3. Line 156, 159, 255, 295: These sentences are fragmented due to typo error or duplicates.
4. Line 164: 16 -> 15 ("12 previously reported SMRT RenSeq and three new assemblies from accessions SP2298, SP3370, SP2308" mentioned on lines 128-129 of the Supplemental Information)
5. Line 174-176: It would be better to more explain why NRC1 was tested although Rpi-amr3 and Rpi-amr1 require NRC2/3/4 and NRC2/3, respectively.
6. Line 413: NLR1 -> NRL1

Reviewer #2:

Remarks to the Author:

Late blight caused by *Phytophthora infestans* was and remains one devastating disease to potato plant, a major staple crop across the world. In this study, the authors generated high-quality genomes of four *Solanum americanum* species that are resistant to late blight and, by combining with 52 resequenced accessions, generated a NLR repertoire in the *S. Americanum* species. Furthermore, the authors generated an ETI landscape of *S. Americanum* accessions against *P. infestans* infection and, by a combination of phenotyping, GWAS and marker-assisted cloning, identified three novel NLRs with the corresponding effector genes. This study represents a remarkable achievement in understanding potato-*Phytophthora infestans* interactions and the valuable genomic information and important genetic tools that are generated/developed could greatly facilitate resistance gene discovery and disease resistance improvement in potato plants.

Main comments:

1. What are the disease phenotypes of late blight on SP2271 wild type and Rpi-amr4ko plants (eg., after inoculation of strain T30-4 or 88069)? Based on results in Supplementary Fig. 15, the two *P. infestans* strains both contain Avramr4. The authors only showed effector recognition phenotype in Fig. 4e, but not the actual disease resistance phenotype. From the description in the main text and methods, I assume the NLR gene is stably knocked out in the mutant plants. The same applies to the nlr16 mutant (Fig. 5).

2. The presentation can be improved to increase clarity:

Fig. 2a, the labels of different plant species/cultivars (I assume, along the blue scale) are too small and not visible.

Labeling on Fig. 4c is not clear.

Fig. 4e, the cell death is not clear and the leaf pictures are too small.

Supplementary Fig. 17f, it's better to quantify the disease symptoms or lesion size.

3. The manuscript should be checked carefully, since there are a few grammar errors. For example, Lines 156-159, lines 254-255, line 295, lines 331-332.

Reviewer #3:

Remarks to the Author:

The manuscript by Xiao et al presents an incredible amount of high quality research to disclose the genetic reservoir of *Solanum Americanum*. The authors provide examples how late blight disease resistance can be mined using this resource. The novelty of this work is that pangenomics from the host is linked directly to effectoromics from the pathogen. The cloning of three receptor encoding genes from complex clusters of paralogous sequences and the simultaneous identification of their ligands shows that the presented approach provides an unprecedented efficiency.

Major points for improvement

The way the expression analysis of NLR and specifically helper NLR's is presented does not really contribute to the general comprehension of the presented work. Could you make a distinction between expression levels of helpers, and that way distinguish them from sensors? Such an addition would increase the relevance of this section.

The transient complementation experiment of Rpi-amr4 in bentha shows partial resistance, while the resistance level in stable transformants seems to provide full resistance. However, in contrast to the transient complementation these data (fig S15) are not quantified. Is that resistance also partial when compared to Rpi-amr3? If so, could late blight resistance in the F2 population be tested for co-segregation with Rpi-amr4? Alternatively, could the resistance in SP2271-knock-out plants be compared to SP2271?

Why is the transient/stable complementation for Rpi-amr16 and 17 not included? Was there no resistance observed in these assays? If Late blight resistance is not shown, than the gene cannot be called Rpi. Rather ERpi. (effector response)

The nomenclature of Rpi allelic variants is indicated with a dot followed by a sequential number (ie Rpi-vnt1.1) Accession names can be added after underscore (ie Rpi-chc1.2_543-5). Active alleles start with a capital. Inactive alleles are written in small font (ie rpi-ber1.4_581-2. It is strongly

recommended that this nomenclature is implemented here as well.

The SMRT-RenSeq assemblies from the 16 different accessions should be included in the bioproject

Reference to Supplementary figures and tables must be improved. Also the legends of the supplementary figures and tables must be improved. Several suggestions are given below.

Specific comments. A quote from the text section (preceded by line number) is followed by comment after >

25-28 Many R 26 genes against *P. infestans* (Rpi genes) were cloned.... 3-10

>reference 3. Park, T.-H. et al. is not the paper that describes the cloning

>R3 is not a gene. It is the name of a family that contains R3a, and R3b

75 based on their variation in resistance to late blight (Supplementary Fig. 1).

> FigS1 does not contain any data about late blight resistance, just growth phenotypes are presented
62 ...novel technologies for potato genome design and engineering, this will help develop potatoes with durable late blight resistance.

> The applications of Rpi-amr genes to engineering potato late blight resistance through recombinant DNA technology is obvious. However, the feasibility of genome design to deploy Rpi-amr genes is rather low regarding number of inversions in 10 chromosomes and the translocations between chromosomes 3, 4, 11 and 12. This point must be addressed

67 ...K-mer based analysis indicated that the genome sizes of these four *S. americanum* accessions were 1.15-1.31 Gb, with 0.05% to 0.35% heterozygosity (Table 1, Supplementary Fig. 2).

> Neither Figure S2 nor Table 1 show how genome size was inferred using k-mer analysis.

79 move "respectively" to line 80

82 The noisy reads were self-corrected

> explain self corrected. the sequence errors were self corrected by the high coverage?

85 SP2273, and anchored the contigs into 12 pseudomolecules (Supplementary Fig. 3)

> Figure S3 shows a quality assessment of the assemblies, not the supercontig compilation as suggested here

125 56 large SVs in SP2271 (Supplementary Fig. 6), impacting ~256 Mb of the reference genome

> Did you validate these are not assembly errors? If available, include the data and make a statement in the text that these SVs are probably no assembly artefacts

135 SVs might contribute to gene expression variation.

> Provide a reference

139 expression might be impaired by a 286 bp deletion in the promoter region in leaves of SP2271

> This is confusing. Do you mean: The reduced expression of gene... in leaves from SP2271 was associated with a deletion in the promoter?

144 As an example, the Rpi-amr3 locus varied greatly among *S. americanum* genomes

(Supplementary Fig. 9b), suggesting the NLRome based on high-quality genomes is required for deep understanding the non-host resistance of *S. americanum* species.

> this statement is not to the point. I agree, however, that chromosome sized assemblies provide insights in NLRome evolution, possibly resulting from co-evolution with pathogens

160 NLR gene models, we manually annotated

> explain somewhere (Fig 2 legend?) what you mean with manually

164 and from 16 SMRT-RenSeq assemblies,

> Explain that these assemblies were derived from 16 different *Solanum* accessions

167 Many well-known NLR genes are relatively highly expressed, such as ADR1, NRG1, Rpi-amr3,

NRC4a, NRC2 and NRC3 (Fig. 2f). Many Solanaceae CNLs require 169 NRC helpers to execute their function and 50% of the *S. americanum* NLRs lie within the NRC superclade 40 170 (Fig. 2a).

> explain NRC superclade. In figure 2a you mention NRC dependent clade. Are these the same?

171 We found NRC1, NRC2, NRC3, NRC4a and NRC6 homologs, two NRC5a copies.

> What is the difference between homologs and copies. Improve this sentence.

172 Interestingly, the copy number of NRC4b homologs has expanded in the *S. americanum* genome.

> provide numbers

173–176 is a confusing section. Please improve. Indicate what is the relevance of this experiment in the pangenome context

188 ... most if not all expressed RXLR effectors were included in this screening

> clarify which plant material was used for this expression analysis. Also from the MPP paper it is not clear if *S. tuberosum* or *S. Americanum* was used

193 AVRamr1 and AVRamr3 are also widely recognized by different *S. americanum* accessions (Fig. 3)

> specify "widely" with numbers

212 -214

> You refer to Fig S12 for the phylogenetic analysis of the *Sa* accessions, while the data are presented in Figure 3. So refer to Figure 3. Only Fig S12c is uniquely referred to in the text (line 220)

215-217

These results were already presented in lines 196-199. So rearrange these two sections.

233-235. From the supplementary methods it is not clear which sequence data were used for the GWAS. Please clarify. Indicate in the text how many accession were included in the GWAS analysis.

237 This likely explains...

> Clarify what you mean with "This". Did the reads from chromosome 1 wrongly map to chr11? This could be confirmed if the chr11 reads that produced the high LOD score have low mapping quality

242 The PITG_22825 responsive gene from SP2271 was mapped to the same position.

> Chr1 or chr 11? The Renseq result that is shown in Fig S 13 is biased towards the candidate gene on Chr1. So provide evidence to support "the same position"

285 ... effector responsive genes ...

> the genes are not responsive to the effector. So consistently use effector response genes

314 ...BC1 population of SP2271(NR) x SP2300 (R) by transiently expressed

> replace by with and

416-434. This section contains wild speculations about many different aspects that only weakly contribute to your conclusion that "non host" resistance can be achieved by R gene stacking. Please revise this section. Is it the sheer number of R genes in a stack that makes it unsurmountable by the simultaneous loss of effectors? Or are R genes from "non-host" plants more effective due to proven durability in co-evolution? Several reports about R gene stacking are available but are not referred to. Also realise that a gene can not be weak (as stated repeatedly). Probably you mean that Rpi gene mediated resistance is weak? Then please define weak. Be careful as HR amplitude (what you are studying in this manuscript mainly) does not necessarily correspond with disease resistance.

439 Conceivably, Rpi-amr4, Rpi amr16 and Rpi-amr17 might also recognize the effectors from other important Phytophthora pathogens and confer resistance against multiple diseases.

– This is just a loose remark. It must either be removed or substantiated to make a decent point.

497 8. Vossen, E. A. G. et al.

>van der Vossen

Suggestions to improve Tables and Figures

Table 1: Title is missing

It is not explained why anchor rate and chromosome number are missing for accession 2275

Figure 2:

Not all abbreviations/gene names in figure 2a and 2b are correct, explained, and/or referred to.

Figure 2a and f: The PDF file provides insufficient resolution to view all information

Figure 2b. Does the length of yellow and red line/dots proportional to the number of genes found?

The position of the gene names should be indicated consistently to the left OR right side of the chromosome

For the comprehension and coherence with Fig 2e and 2f it would be good to include the position of the large NRC4b-like clade, NRC5 and NRC 6 as well

Figure 2e the *S. americanum* NRC orthologs should be named Sa instead of SP1102, to be consistent with the naming of the other species

Figure 2d. Indicate which color corresponds to manually curated or automatically predicted

Figure 3: specify the transient expression system that was used

Figure 4: 4c What is HF? The text mentions Flag

4d: appears after figure 4f. So change order. The y axis values (lesion size?) must be added. Rpi-amr4_1102 must be mentioned in the legend.

Figure S2: The title should read: k-mer frequency distribution in *S. americanum* genome reads

Figure S4: indicate which BUSCO database has been used

Figure S5: indicate which window size was used

Figure S6: indicate which reference genome was used

Figure S9b: The different box colors must be explained. The identity threshold for the connecting lines should be given. Did you use a specific software tool to generate this image?

Figure S10: Explain abbreviations. Pp: *Phytophthora palmivora*? Why didn't you use Pi? Explain in the legend

Figure S12: Specify a, b, and c

The legend is insufficiently describing the figure. Briefly describe the data and methods used for the phylogeny, the population structure (color scheme explanation), and inbreeding co-efficient analysis.

Figure S13: separate legends for b and

Provide a definition of informative SNPs.

mention the reference genome

improve the column headers in table C so that it is clear which frequency is meant. frequency of the Alt allele in NR bulk?

Reviewer #4:

Remarks to the Author:

This study has deployed wild accessions of *Solanum americanum* to clone novel immune receptors that recognize *Phytophthora infestans* RXLR effectors. *S. americanum* is resistant to *P. infestans*, yet can be used as source for engineering late blight (caused by *P. infestans*) resistance in potato genotypes used for production. Four *S. americanum* reference genomes were generated that were used to identify novel R gene specificities in 52 accessions, each challenged with 315 *P. infestans* effectors. This matrix search identified a total of three new R gene specificities. The novelty and brilliance of this study lies in the use of a non-host plant to tap on new R genes, an approach that is much more efficient when diploid wild accessions are used rather than higher cultivated genotypes with often higher ploidy levels. I rate this a breakthrough study in R gene identification which, as the authors correctly state, could be applied to any crops and any diseases. Identification of three novel R genes provided a nice proof of principle. The genes identified behave, however, just as any other R

genes (with limited novel insight into R gene function).

the grammar in sentences lines 55 and 254/255 is incorrect

Author Rebuttal to Initial comments

Response to referees:

Reviewer #1:

Remarks to the Author:

Lin et al. described the genetic variation and characterization of NLR immune receptor genes of *Solanum americanum* by constructing a pan-genome of *S. americanum*, and performing an effectoromics screening against 315 *Phytophthora infestans* RXLR effectors in 52 *S. americanum* accessions. Based on resistant phenotype of *S. americanum* accessions against potato late blight, the authors selected and assembled four high-quality de novo genome assemblies. Re-sequencing of 52 *S. americanum* accessions with SMRT-RenSeq revealed phylogenetic relationship, population structure, and the genetic variation of NLRs. Screening of 315 *P. infestans* RXLR effectors in 52 *S. americanum* accessions elucidated the effector-triggered immunity landscape between *S. americanum* and *P. infestans*. Using GWAS, BSA-RenSeq, SMRT-RenSeq, and map-based cloning strategies, three novel Rpi-Avr gene pairs were characterized. Overall, these results and resources are comprehensive and should be very informative to the community. However, there are some major and minor concerns that should be resolved before consider publication of the manuscript.

Major concerns

1. Although the title includes the word “pangenome”, these genomic information does not cover whole genetic variation of *S. americanum* species. The phylogeny and population structure of 52 *S. americanum* accessions separated into four groups (Supplementary Fig. 12a). However, the authors selected four accessions in group 1 to 3 to construct pan-genome. These accessions do not cover the genetic variation of its own group and group 4 (for example, population structure of SP2298 to SP2302 in group 3 and all accessions in group 4). If the “pangenome” mean “pan-NLRome”, the title should be changed. If “pan-NLRome” is what the authors intended to, the saturation curves for core- and pan-NLR numbers to confirm that the pan-NLRome covers the entire NLR repertoire of *S. americanum* should be addressed.

Thanks for the comments, we agree with the reviewer that the 4 *S. americanum* genomes do not cover the whole genetic variation of *S. americanum* species. We have changed the title to

“*Solanum americanum* genome-assisted discovery of new immune receptors that detect potato late blight effectors”. To improve the pan-NLRome analysis, we have generated saturation curves for the *NLR* orthogroups; see the newly added Supplementary Fig 9a. The number of *NLR* orthogroups increased when we added more accessions and nearly reached a plateau when $n = 19$, which suggests that our pan-NLRome captures the entire *NLR* repertoire of *S. americanum*. We also visualized the core, dispensable and unique *NLR* genes from each accession (Supplementary Fig 9b). We added this figure and the corresponding description to the revised manuscript, see line 185-191.

Core Dispensable Unique

Supplementary Figure 9. Pan-NLRome of *S. americanum*. (a). Simulation of pan- and core-NLRome size of *S. americanum*. The NLR orthogroups were classified by OrthoFinder. The number in x axis indicates the number of *S. americanum* accessions were randomly selected for pan- and core-NLR analysis. For each accession number, 500 times of random selection with 30 replicates were performed. The estimated pan- and core-NLRome were fitted with exponential models. (b). Core, dispensable, and unique *NLR* genes from the 20 *S. americanum* accessions. Core *NLR* genes: *NLRs* present in all 18-20 accessions; Dispensable: *NLRs* were missed in more than 3 accessions and present in at least two accessions; Unique: *NLRs* present in only one accession.

2. The authors selected four *S. americanum* accessions based on phenotypic variation in resistance to late blight (Line 75 and Supplementary Fig. 1) but the Supplementary Fig. 1 do not show resistance phenotype. I recommend to include in detail description and phenotype images in the figure.

Thanks for the suggestions, the resistance/susceptible phenotype of SP2271 and SP2273 were included in a previous publication (Witek et al, Nature Plants, 2021, Figure S1.). To improve this manuscript, we have repeated the disease test by one *P. infestans* isolate (T30-4) for all the four reference accessions. Also, to address Reviewer 1's concern 3, we included different *S. americanum* leaves in the disease assay, and potato and *N. benthamiana* were included as controls, see the Supplementary Figure S1b.

3. As the authors described (line 197-198), some accessions of *S. americanum* are susceptible to *P. infestans* (I couldn't understand meanings of the sentence "yet resistant to late blight in the field", Is it by inoculation? or in the natural condition?). Moreover, Witek et al. (2021, Nature Plants) also described the accession SP2271 as a susceptible host of *P. infestans* but the authors described *S. americanum*-*P. infestans* interaction as a nonhost pathosystem (in Line 13, 61, and 433). However, nonhost resistance is often defined as "a resistance of an entire species of plant against an entire species of pathogen". Thus, *S. americanum* in this study could not fit the case of "nonhost resistance" against *P. infestans*. This should be considered and resented in the text.

SP2271 is susceptible to *P. infestans* in lab conditions (detached leaf assays of 4-5 weeks old plants, 3-5 leaves from top, zoospores inoculation), and it was used as the susceptible parent for most our mapping populations. The two mapping populations SP2271 x SP1102 and SP2271 x SP2273 led to the cloning of *Rpi-amr3* and *Rpi-amr1* from SP1102 and SP2273 respectively ¹.

To evaluate the susceptibility of SP2271 in the field conditions, we grew SP2271 next to a potato field in which the susceptible potato genotypes were destroyed by late blight, but we never observed infected SP2271 plants. Therefore, we concluded that adult SP2271 plants resist late blight in field conditions. See the photo below (For reviewing purpose only).

To further address this question, we performed detached leaf assays of different leaves from a same plant, and found that the susceptibility of SP2271 is conditional. As shown in the Supplementary Figure 1b, we tested 6 leaves/ plant (~7-8 weeks old plants grown in containment glasshouse) from top to bottom, all leaves from SP1102, SP2273 and SP2275

are highly resistant to *P. infestans*. For SP2271, the first three leaves are relatively resistant, but the older leaves are more susceptible. We also included wild-type *N. benthamiana* and a potato cultivar as controls, and verified that both *N. benthamiana* and potato are highly susceptible to *P. infestans*, regardless of the leaf age. It is worth noting that even the most susceptible SP2271 leaves are more resistant than the *N. benthamiana* and potato leaves (Supplementary Figure 1b). These results indicate that SP2271 carries some *Rpi* genes that confer quantitative and conditional late blight resistance, consistent with our finding that SP2271 carries functional *Rpi-amr4*, *R02860* and can recognize ~20 additional RXLR effectors. Dissecting and cloning of these immune receptors from SP2271 will let us understand its quantitative and conditional resistance against *P. infestans* in the field.

Supplementary Figure 1b

We agree with the reviewer that it's not convincing to conclude *S. americanum* as a nonhost plants because of the susceptibility of SP2271 in the lab condition. The concept and definition of “nonhost” resistance is still under debate². The phenotype of nonhost resistance is sometimes a spectrum instead of black and white phenotype. However, we believe that our study extends people's understanding on the host/nonhost resistance. In the case of *S. americanum*, most accessions carry functional *Rpi-amr1*, *Rpi-amr3* and many other effector recognitions, the less resistant accessions like SP2271 lack the functional *Rpi-amr1/3*, but carry additional immune receptors like *Rpi-amr4* and R02860 (As suggested by Reviewer 3, We renamed *Rpi-amr16* as R02860), and ~20 additional recognitions. We hypothesise that these effector recognitions might contribute to its quantitative and age-dependent resistance against late blight. The genomic and genetic resources we have generated in this study will help to clone more immune receptors from SP2271 and other accessions.

To avoid further confusion, we now avoid using the term “nonhost resistance” in the revised manuscript.

4. Among the newly identified three NLRs (*Rpi-amr4/16/17*), the authors showed only the case of *Rpi-amr4* exhibited partial/quantitative resistance against *P. infestans* and clearly

confirmed the relationship between resistance phenotype of *S. americanum* against *P. infestans* and Rpi-amr4 using knockout lines and complementation assay. However, for the cases of Rpi-amr16 and 17, the authors presented each NLRs could induce HR-like cell death against corresponding effectors without any evidence on the blight resistance in planta. Quite a few of those cases are not end-up with actual disease resistance in plants. Thus, at least, transient expression-mediated resistance against *P. infestans* should be presented using Rpi- amr16/17 (because of the autoactive nature of those NLR with strong promoter such as 35S, authors may need its native promoter).

Thanks for the suggestions. To address this question, we cloned the R02860 (Rpi-amr16) with its native promoter and terminator; this construct is not auto-active when transiently expressed in *N. benthamiana*. We evaluated these constructs in *N. benthamiana* transiently, see the newly added Supplementary Figure 19. However, our data indicate that the effects of native R02860 (Rpi-amr16) and 35S::R04373 (Rpi-amr17) on *P. infestans* resistance are slight, although statistical analysis indicates that there is a significant difference in lesion size when R02860 (Rpi-amr16) and R04373 (Rpi-amr17) are compared with the negative control (Rpi-amr1-2271, an non-functional *Rpi-amr1* homolog from SP2271). We don't think these slight differences of lesion size are practically useful to control potato late blight in the field.

Supplementary Figure 19

The HR phenotype is in a rather artificial setting, upon over-expression of pathogen effectors without infection. However, during the natural infection, the pathogen can overcome the HR by multiple strategies, for example, silencing the effectors or suppression of plant immunity. In our case, the R02860 (Rpi-amr16) mediated HR can only be observed when over-expressing the effector using PVX, but not with the 35S promoter. This indicates that R02860 can only be activated when encountering massive amounts of PITG_02860 protein. For R04373 (Rpi-amr17), we have performed a suppressor screening for loss of HR phenotype and found one RXLR effector that can suppress R04373 mediated HR in *N. benthamiana*

(See the figure below, for reviewing purpose only), this indicates that a complex effector-receptor defence and counter defence network that beyond the gene-for-gene model. We think these data are beyond the scope of the current manuscript, and we hope to report these data in a separate paper. Although R02860 (Rpi-amr16) and R04373 (Rpi-amr17) do not confer strong late blight resistance, we believe these findings will help to understanding the arms race of *P. infestans* and plants.

Supporting figure for reviewing purpose only.

5. Data availability: There are missing datasets. The manual annotation of NLRs was performed to SP1102, SP2271, and SP2273, but only SP1102 was provided (Supplementary Dataset 1 to 3). Resequencing data of 52 *S. americanum* accessions and SMRT RenSeq data of SP2298, SP3370, and SP2308 also missed. Furthermore, I strongly recommend uploading of the variant calling file (VCF) to Figshare.

Thanks for the suggestion, the manually annotated *NLR* genes and proteins for SP2271 and SP2273 were added to Supplementary data 1-3.

The resequencing data of the 52 *S. americanum* accessions have been deposited to ENA, under project ID: PRJEB57057 (ERP142025)

We have deposited all the SMRT RenSeq assemblies and the annotation (NLR-annotator) in FigShare:

https://figshare.com/articles/dataset/Solanum_americanum_SMRT_RenSeq_Assemblies/21657017

We have uploaded the VCF files (raw VCF files and two filtered VCF files that used for the phylogeny and GWAS analysis respectively, as well as the VCF files for structural variations) to Figshare:

https://figshare.com/articles/dataset/VCF_files_for_the_phylogenetic_structural_and_GWAS_analysis/21647057

https://figshare.com/articles/dataset/Variation_information_of_S_americanum/21441567

Minor Concerns

1. Supplementary Fig. 6: It would be better to state or describe that the reference is SP1102.

Added.

2. Line 145-147: The paragraph described the diversity of NLR region compared to non-NLR region. This is thought to be more related to the host resistance by NLR diversity than to the “nonhost” resistance.

Most *S. americanum* accessions are not hosts of *P. infestans*, but they can be hosts of other plant pathogens. Therefore, we think it's not a surprise that *NLR* genes show a higher polymorphism than *non-NLR* genes. But as discussed above, we agree that the concept of nonhost resistance is ambiguous, so we now avoid using the term in the revised manuscript.

3. Line 156, 159, 255, 295: These sentences are fragmented due to typo error or duplicates.

We have corrected the sentences.

4. Line 164: 16 -> 15 (“12 previously reported SMRT RenSeq and three new assemblies from accessions SP2298, SP3370, SP2308” mentioned on lines 128-129 of the Supplemental Information)

Thanks for the reviewer’s sharp eyes, in total we have 16 SMRT RenSeq assemblies, we corrected the sentence to: “13 previously reported SMRT RenSeq and 3 new assemblies”

5. Line 174-176: It would be better to more explain why NRC1 was tested although Rpi-amr3 and Rpi-amr1 require NRC2/3/4 and NRC2/3, respectively.

We specified that “Rpi-amr3 and Rpi-amr1 require NbNRC2/3/4 and NbNRC2/3 in *N. benthamiana*, respectively ^{1,3}. However, *NRC1* is missing in *N. benthamiana* ⁴. To test if *NRC1* from *S. americanum* can enable *Rpi-amr1/3* function, we cloned the *SaNRC1* from SP1102 and showed *SaNRC1* enables *Rpi-amr3* but not *Rpi-amr1* function in *N. benthamiana* *nrc2/3/4* knockout line”

6. Line 413: NLR1 -> NRL1

Corrected.

Reviewer #2:

Remarks to the Author:

Late blight caused by *Phytophthora infestans* was and remains one devastating disease to potato plant, a major staple crop across the world. In this study, the authors generated high-quality genomes of four *Solanum americanum* species that are resistant to late blight and, by combining with 52 re-sequenced accessions, generated a NLR repertoire in the *S. Americanum* species. Furthermore, the authors generated an ETI landscape of *S. Americanum* accessions against *P. infestans* infection and, by a combination of phenotyping, GWAS and marker-assisted cloning, identified three novel NLRs with the corresponding effector genes. This study represents a remarkable achievement in understanding potato- *Phytophthora infestans* interactions and the valuable genomic information and important genetic tools that are generated/developed could greatly facilitate resistance gene discovery and disease resistance

improvement in potato plants.

Main comments:

1. What are the disease phenotypes of late blight on SP2271 wild type and Rpi-amr4ko plants (eg., after inoculation of strain T30-4 or 88069)? Based on results in Supplementary Fig. 15, the two *P. infestans* strains both contain Avr_{amr4}. The authors only showed effector recognition phenotype in Fig. 4e, but not the actual disease resistance phenotype. From the description in the main text and methods, I assume the NLR gene is stably knocked out in the mutant plants. The same applies to the nlr16 mutant (Fig. 5).

As also suggested by other reviewers, we tested the late blight resistance of R02860 (Rpi-amr16) and R04373 (Rpi-amr17) in the *N. benthamiana* transient expression system. Our data indicate that the resistances conferred by native R02860 (Rpi-amr16) and 35S::R04373 (Rpi-amr17) are weak, although the statistical analysis indicates that there is a significant difference of the lesion size when the the negative control Rpi-amr1-2271 is compared with R02860 (Rpi-amr16) and R04373 (Rpi-amr17).

An *Rpi-amr4* allele is also present in SP2271, but Rpi-amr4-2271 is auto-active in *N. benthamiana* (Fig. 4c). Therefore, we only tested the Rpi-amr4-1102 and concluded it confers quantitative resistance. Similarly, the R02860 (Rpi-amr16) from SP2271 is auto-active under 35S promoter. To test the efficacy of R02860 from SP2271, we managed to clone the full-length gene with its native promoter and terminator and this construct is not auto-active in *N. benthamiana*. However, the native Rpi-amr16-2271 does not confer strong late blight resistance, see the newly added Supplementary Figure 19.

We also generated *Rpi-amr4*, *R02860* (*Rpi-amr16*) and *Rpi-amr4/R02860* double knockout SP2271 lines and confirmed these lines by genotyping and agro-infiltration of the corresponding effectors (Figure 4e and Figure S17). For the *Rpi-amr4/R02860* double ko SP2271 lines (not included in the MS), and also verified these lines by genotyping and agroinfiltration, see the photo below (For review purpose only). We generated T1 seeds from all the three knockout lines, and performed agroinfiltration again, the results indicate that all the gene editing events can be stably inherited.

To test if the *Rpi-amr4*, *R02860* (*Rpi-amr16*) and *Rpi-amr4/R02860* knockout lines are more susceptible to *P. infestans*, we firstly tested the young wild-type SP2271 plants and the knockout lines (~4 weeks). However, we didn't observe any differences in lesion size between the knockout lines and the wild-type SP2271 (T30-4, 5 dpi).

As shown above (see response to reviewer 1, and newly added Supplementary Figure 1b), the susceptibility of SP2271 is age-dependent. Therefore, we further tested different leaves from older plants (~7-8 weeks) and performed ANOVA test for all the leaves, leaf 1-3 and leaf 4-6, but we didn't observe significant differences. See the data below:

7-8 weeks plants, leaf 1-6, T30-4, 50,000/mL. 10 uL, 6 dpi

7-8 weeks plants, leaf 1-3, T30-4, 50,000/mL. 10 uL, 6 dpi

7-8 weeks plants, leaf 4-6, T30-4, 50,000/mL. 10 uL, 6 dpi

In summary, although we verified that the effector recognition capacity of *Rpi-amr4*-2271 and *Rpi-amr16*, and we showed that *Rpi-amr4*-1102 confers a quantitative late blight resistance. However, we didn't observe an increased susceptibility in the *Rpi-amr4*-2271, *R02860* and *Rpi-amr4/R02860* knockout SP2271 lines. As shown in Figure S11, SP2271 recognizes ~20 RXLR effectors from *P. infestans*, knocking out two recognitions might not be sufficient to make SP2271 more susceptible. Using the genomic and genetic resources generated in this study, these resources will enable the cloning of more immune receptors from SP2271 and dissect its quantitative and conditional resistance against late blight.

For the revision, we included the transient disease assay in *N. benthamiana* for *R02860* (*Rpi-amr16*) and *R04373* (*Rpi-amr17*) (newly added Supplementary Figure 19), renamed the genes from *Rpi-amr16/17* to *R02860/R04373* (receptor of PITG_02860 and PITG_04373) as suggested by Reviewer 3, and we made the point that both genes do not confer strong late blight resistance. We think that including the disease assay results for the *Rpi-amr4*, *R02860* and *Rpi-amr4/R02860* lines will be redundant and unnecessary, and we plan to perform field disease test for these knockout lines in the future, to see if these genes involve the field resistance against late blight of SP2271. Therefore, the disease test data of these knockout lines is not included in the revised manuscript.

2. The presentation can be improved to increase clarity:

Fig. 2a, the labels of different plant species/cultivars (I assume, along the blue scale) are too small and not visible.

We are not able to further increase the font size, this information can be found in Table S2, and we added the name in the legend of Fig. 2.

Labeling on Fig. 4c is not clear.

Fig. 4e, the cell death is not clear and the leaf pictures are too small. We

have adjusted Figure 4 as suggested.

Supplementary Fig. 17f, it's better to quantify the disease symptoms or lesion size.

In Figure S18.f, we showed disease symptoms of two *rpi-amr1/3/r04373* T0 lines (SLJ25603#3 and SLJ25603#17). To quantify the lesion size of the two ko lines, we repeated this experiment on the T1 lines that derived from the two T0 lines. We first transiently expressed *AVRamr1*, *AVRamr3* and *PITG_04373* in the T1 leaves. Same as the T0 lines, all the effector recognitions were gone. Then we performed DLA, same as the T0 lines, the two

T1 lines all show a very small lesion after *P. infestans* T30-4 inoculation. The lesion size is bigger than the wild-type SP2300 (See the newly added Supplementary Figure 18g). These data indicate that there are more *Rpi* genes in SP2300 in addition to *Rpi-amr1*, *Rpi-amr3* and *R04373*. These triple knockout lines would be ideal starting materials for cloning more novel *Rpi-amr* genes, and our genomic resources and effectoromics screening will accelerate the new gene cloning.

3. The manuscript should be checked carefully, since there are a few grammar errors. For example, Lines 156-159, lines 254-255, line 295, lines 331-332.

All corrected.

Reviewer #3:

Remarks to the Author:

The manuscript by Xiao et al presents an incredible amount of high quality research to disclose the genetic reservoir of *Solanum Americanum*. The authors provide examples how late blight disease resistance can be mined using this resource. The novelty of this work is that pangenomics from the host is linked directly to effectoromics from the pathogen. The cloning of three receptor encoding genes from complex clusters of paralogous sequences and the simultaneous identification of their ligands shows that the presented approach provides an unprecedented efficiency.

Major points for improvement

The way the expression analysis of NLR and specifically helper NLR's is presented does not really contribute to the general comprehension of the presented work. Could you make a distinction between expression levels of helpers, and that way distinguish them from sensors? Such an addition would increase the relevance of this section.

We have noted all the known helper and sensor *NLR* genes from SP1102 genome in the list of expressed genes, see the updated Table S1. Although most helper NLRs are highly expressed in our cDNA-RenSeq dataset, we can't distinguish the sensor/helper NLRs based on the expression level. We want to emphasise that the cDNA data is a very useful criterion for predicting the right candidate gene after the BSA-RenSeq. Firstly, we could rule out all the non-expressed *NLRs*, and we found the most highly expressed NLR genes are typically the right candidate gene, for example, both *Rpi-amr3* and *Rpi-amr4* are highly expressed in SP1102. In our previous work, *Rpi-amr1e* (*Rpi-amr1*) is also the most highly expressed *NLR* in the *Rpi-amr1* cluster of SP2273¹.

The transient complementation experiment of Rpi-amr4 in bentha shows partial resistance, while the resistance level in stable transformants seems to provide full resistance. However, in contrast to the transient complementation these data (fig S15) are not quantified. Is that resistance also partial when compared to Rpi-amr3? If so, could late blight resistance in the F2 population be tested for co-segregation with Rpi-amr4? Alternatively, could the resistance in SP2271-knock-out plants be compared to SP2271?

Yes, the transient disease assay of 35S::*Rpi-amr4-1102* in *N. benthamiana* showed partial resistance, and the stable 35S::*Rpi-amr4-1102* transgenic *N. benthamiana* (T0) lines showed full resistance, so we didn't quantify the symptom.

Now we generated the T1 seeds from the T0 35S::*Rpi-amr4-1102* transgenic *N. benthamiana* and we have repeated the disease assay on the T1 lines. The T1 lines are segregating to the AVR_{amr4} responsiveness, therefore we firstly transiently expressed AVR_{amr4} in the T1 lines, and the AVR_{amr4} responsive lines were selected for the DLA. Similarly, most of the AVR_{amr4} responsive T1 lines are completely resistant to *P. infestans* T30-4, with a few exceptions, see the updated Supplementary Figure 16b below:

It's difficult to test the resistance of *Rpi-amr4* in the *S. americanum* population, because SP2271 is the “susceptible” parent for most of our crossings, and SP2271 also carries an *Rpi-amr4* homolog that recognizes AVR_{Amr4}. The efficacy of *Rpi-amr4-2271* is not easy to detect in the segregating population. Unfortunately, the *Rpi-amr4-2271* allele is auto-active in *N. benthamiana*, therefore we could not further test for late blight resistance. And we didn't observe an increased susceptibility in the *Rpi-amr4* knockout SP2271 (see response to Reviewer 2 about the *Rpi-amr4*, *R02860* and *Rpi-amr4/R02860* knockout SP2271 lines). For *Rpi-amr4-1102*, we used the p35S promoter for both transient assay and stable transformation, the strong promoter might enhance its resistance. To compare the 35S promoter and the native promoter, we have cloned the *Rpi-amr4-1102* under its native promoter and terminator, then performed the transient disease assay in *N. benthamiana*. The 35S::*Rpi-amr4* confers a stronger late blight resistance compared to native *Rpi-amr4*, see the figure below (for reviewing purpose only).

These results have also puzzled us for a while, but when we revisited the “susceptibility” of SP2271 (see the updated Supplementary Figure 1b), and with the effectoromics screening data, we now know the late blight resistance in *S. americanum* is more complex than we thought. We summarize our current understandings about late blight resistance in *S. americanum* as follows:

1. Even the most susceptible accession SP2271 is more resistant than *N. benthamiana* and potato cultivars, see the updated Supplementary Figure 1b.
2. The “susceptibility” of SP2271 is age and environment dependent. The young leaves

- from older plants are more resistant to late blight, see the updated Supplementary Figure 1b. SP2271 is resistant to late blight in field (see the response to Reviewer 1).
3. SP2271 can recognize ~20 RXLR effectors, and these effector recognitions might contribute to its late blight resistance.

Taken together, the resistance of a plant species cannot simply fit into host/nonhost or resistant/susceptible catalogues. In our case, most *S. americanum* accessions carry *Rpi-amr1*, *Rpi-amr3* or both, therefore they cannot be infected by *P. infestans*, in addition, most *S. americanum* accessions can recognize > ~20 RXLR effectors, which might contribute to their quantitative resistance against *P. infestans*.

Why is the transient/stable complementation for Rpi-amr16 and 17 not included? Was there no resistance observed in these assays? If Late blight resistance is not shown, than the gene cannot be called Rpi. Rather ERpi. (effector response)

We have renamed Rpi-amr16 and Rpi-amr17 to R02860 and R04373 (receptor of PITG_02860 and PITG_04373).

p35S::R02860 (Rpi-amr16) is auto-active in *N. benthamiana*, therefore we didn't test it previously. As suggested by Reviewer 1, we made new R02860 construct with its own promoter and terminator, this construct is not auto-active in *N. benthamiana*, therefore we performed a disease test for native R02860 and 35S::R04373 after transient expression in *N. benthamiana*, see the newly added Figure 19 below:

In this assay, native *Rpi-amr1-2271* (an unfunctional homolog of *Rpi-amr1*) was used as the negative control, native *Rpi-amr3* was used as a positive control. We also included 35S::*Rpi-amr4*, and co-express *R02860* and *R04373* with *Rpi-amr4*. These results indicate that native *R02860* and 35S::*R04373* can slightly slow down the pathogen growth, the lesion sizes are statistically different when compared to the control. However, the difference is small. We agree with reviewer#3's suggestions, it might be misleading if we call them *Rpi* genes if they don't confer strong late blight resistance, therefore, we decided to rename them as *R02860* and *R0437*. Also, we have changed the title to "*Solanum americanum* genome-assisted discovery of new immune receptors that detect potato late blight effectors"

It is not the first example that HR phenotype in an overexpression setting does not correlate with pathogen resistance, for example, the potato cultivar Sarpò Mira is highly resistant to most *P. infestans* isolates, it recognized many RXLR effectors, but carries less dominant *Rpi* genes⁶. But the mechanism behind this phenotype has not been well investigated. Recently, our colleagues reported some pathogen effectors, including the AVRcap1b from *P. infestans* could suppress ETI by targeting the helper NLR NRCs⁷. Here, *R02860* (*Rpi-amr16*) and *R04373* (*Rpi-amr17*) are not NRC dependent, finding their suppressors would help us to further understand the defence/ counter defence of *P. infestans* and plants. To test this, we performed a suppressor screening for the potential suppressors of *R02860* and *R04373*, we've already successfully identified a *P. infestans* RXLR effector which can efficiently inhibit *R04373* mediated HR but does not suppress the NRC dependent NLRs like *Rpi-amr4*/AVR*amr4*.

However, we think this data is out of scope of the current manuscript, therefore we plan to screen more effectors, investigate their mechanism, and report these data in a separate manuscript.

The nomenclature of Rpi allelic variants is indicated with a dot followed by a sequential number (ie Rpi-vnt1.1) Accession names can be added after underscore (ie Rpi-chc1.2_543-5). Active alleles start with a capital. Inactive alleles are written in small font (ie rpi-ber1.4_581-2). It is strongly recommended that this nomenclature is implemented here as well.

We used a slightly different nomenclature system in our previous publications, like Rpi- amr3a, Rpi-amr3i, Rpi-amr1-2273, Rpi-amr1-2272 etc. We hope to keep this nomenclature to keep it more consistent.

The SMRT-RenSeq assemblies from the 16 different accessions should be included in the bioproject

We have uploaded all the SMRT-RenSeq assemblies and the annotations (by NLR-annotator) to Figshare, see the link below:

https://figshare.com/articles/dataset/Solanum_americanum_SMRT_RenSeq_Assemblies/21657017

Reference to Supplementary figures and tables must be improved. Also the legends of the supplementary figures and tables must be improved. Several suggestions are given below.

Specific comments. A quote from the text section (preceded by line number) is followed by comment after >

25-28 Many R 26 genes against *P. infestans* (Rpi genes) were cloned.....3-10
>reference 3. Park, T.-H. et al. is not the paper that describes the cloning **We**

updated the citation (Lokossou et al., 2009)

>R3 is not a gene. It is the name a family that contains R3a, and R3b

Corrected.

75 based on their variation in resistance to late blight (Supplementary Fig. 1).

> FigS1 does not contain any data about late blight resistance, just growth phenotypes are presented

We updated Figure S1, see Figure S1b.

62..... novel technologies for potato genome design and engineering, this will help develop potatoes with durable late blight resistance.

> The applications of Rpi-amr genes to engineering potato late blight resistance through recombinant DNA technology is obvious. However, the feasibility of genome design to deploy Rpi-amr genes is rather low regarding number of inversions in 10 chromosomes and the translocations between chromosomes 3, 4, 11 and 12. This point must be addressed.

We agree that it's not feasible to introgress the *Rpi-amr* genes by crossing, so far, our attempts of crossing *S. americanum* and *S. tuberosum* have not succeeded. We rephrased this sentence to: "In combination with novel technologies like gene knock-in and potato genome design ⁸, this will help to develop better potato varieties with durable late blight resistance."

67..... K-mer based analysis indicated that the genome sizes of these four *S. americanum* accessions were 1.15-1.31 Gb, with 0.05% to 0.35% heterozygosity (Table 1, Supplementary

Fig. 2).

> Neither Figure S2 nor Table 1 show how genome size was inferred using k-mer analysis.

The estimated genome size = total number of k-mers / estimated sequencing coverage, the total number of k-mers could be calculated from sequencing data and sequencing coverage could be assessed based on k-mer distribution frequency. In this study, we applied KAT to calculate k-mer frequency with $k = 19$ and a Perl script `estimate_genome_size.pl` (https://github.com/josephryan/estimate_genome_size.pl) to estimate the genome size of *S. americanum*. We also added the description of this approach into Methods (line 22 - 28 in Supplementary information.).

79 move “respectively” to line 80 **Corrected**

82 The noisy reads were self-corrected

> explain self corrected. the sequence errors were self corrected by the high coverage?

Due to the relatively high error rate of the ONT platform, which will challenge genome assembly, we adopted the 'correct' method embedded in the Canu software to improve the accuracy of long reads. The correction is based on the reads all-versus-all self-overlap information. Except for reads coverage that will be considered, the authors also introduced several algorithms to filter repeat reads and optimize the correction strategy. This information was mentioned in the materials and methods section, see line 30-35.

85 SP2273, and anchored the contigs into 12 pseudomolecules (Supplementary Fig. 3)

> Figure S3 shows a quality assessment of the assemblies, not the supercontig compilation as suggested here

We have updated Figure S3 with supercontig compilation information, the interaction maps were visualized by Juicebox. To display the whole genome interaction, the resolution was set to 2.5 Mb.

125 56 large SVs in SP2271 (Supplementary Fig. 6), impacting ~256 Mb of the reference genome

> Did you validate these are not assembly errors? If available, include the data and make a statement in the text that these SVs are probably no assembly artefacts

Most of the SVs reside in single contigs, with only two exceptions, and 40 of the SVs were supported by Hi-C map, see the updated Figure S6b and newly added Table S1. We also added the statement to the main text (line 130-132) and Supplementary information (line 96- 99).

135 SVs might contribute to gene expression variation.

> Provide a reference

We have added the reference "Major Impacts of Widespread Structural Variation on Gene Expression and Crop Improvement in Tomato" to this sentence.

139 expression might be impaired by a 286 bp deletion in the promoter region in leaves of SP2271

> This is confusing. Do you mean: The reduced expression of gene... in leaves from SP2271 was associated with a deletion in the promotor?

Yes, we rephrased this sentence to "For example, the reduced expression of *sp1102chr11_nlr_6* in leaves from SP2271 might associated with the 286 bp deletion in its promoter region."

144 As an example, the Rpi-amr3 locus varied greatly among *S. americanum* genomes (Supplementary Fig. 9b), suggesting the NLRome based on high-quality genomes is required for deep understanding the non-host resistance of *S. americanum* species.

> this statement is not to the point. I agree, however, that chromosome sized assemblies provide insights in NLRome evolution, possibly resulting from co-evolution with pathogens

Thanks for the suggestion, we rephrased this sentence, and moved it to the NLR section, see line 185-191.

160 NLR gene models, we manually annotated

> explain somewhere (Fig 2 legend?) what you mean with manually

We described the manual annotation pipeline in the materials and methods, see line 120-129 in the Supplementary information.

164 and from 16 SMRT-RenSeq assemblies,

> Explain that these assemblies were derived from 16 different *Solanum* accessions

Done, see line 159, and all the SMRT RenSeq contigs and annotations have been uploaded to Figshare:

https://figshare.com/articles/dataset/Solanum_americanum_SMRT_RenSeq_Assemblies/21657017

167 Many well-known NLR genes are relatively highly expressed, such as ADR1, NRG1, Rpi-amr3, NRC4a, NRC2 and NRC3 (Fig. 2f). Many Solanaceae CNLs require 169 NRC helpers to execute their function and 50% of the *S. americanum* NLRs lie within the NRC superclade 40 170 (Fig. 2a).

> explain NRC superclade. In figure 2a you mention NRC dependent clade. Are these the same?

The NLR superclade was described in (Wu et al., 2017), we cited the reference, and to clarify this, we rephrased the sentence to:

“Many Solanaceae CNLs require helper NLR NRCs that are phylogenetically related and grouped into an NRC superclade. We found that about 50% of the *S. americanum* NLRs lie within the NRC superclade”

For figure 2a, we changed NRC dependent clade to NRC superclade in both figure and figure legend.

171 We found NRC1, NRC2, NRC3, NRC4a and NRC6 homologs, two NRC5a copies.

> What is the difference between homologs and copies. Improve this sentence.

We removed copies and use homologs, see line 175.

172 Interestingly, the copy number of NRC4b homologs has expanded in the *S. americanum* genome.

> provide numbers

Provided, see line 176-177:

NRC4b genes (7 homologs) have expanded in the *S. americanum* genome compared to *N. benthamiana* (2 homologs).

173—176 is a confusing section. Please improve. Indicate what is the relevance of this experiment in the pangenome context

Previously, most of the NRC dependency data were produced in the *nrc* knockout *N. benthamiana* lines, however NRC1 is missing in *N. benthamiana*. Here we cloned SaNRC1 and showed SaNRC1 can also support the function of Rpi-amr3, these results connected the in-silico assays of the NLRome and their function. And this data expand our knowledge on

the NRC network in different plant species.

We have improved this section to better convey our message.

188 ... most if not all expressed RXLR effectors were included in this screening

> clarify which plant material was used for this expression analysis. Also from the MPP paper it is not clear if *S. tuberosum* or *S. Americanum* was used

We rephrased the sentence to: Based on the cDNA PenSeq data, all these RXLR effectors are expressed upon infection of potato.

Thanks for pointing out that this information is ambiguous in the MPP paper, we mentioned potato in the experimental procedures- sample preparation section, we should have specified this more clearly in the results section.

193 AVRamr1 and AVRamr3 are also widely recognized by different *S. americanum* accessions (Fig. 3)

> specify “widely” with numbers

The number was added, see line 208-209: AVRamr1 (36/52) and AVRamr3 (43/52) are also widely recognized by different *S. americanum* accessions

212 -214

> You refer to Fig S12 for the phylogenetic analysis of the *Sa* accessions, while the data are presented in Figure 3. So refer to Figure 3. Only Fig S12c is uniquely referred to in the text (line 220)

We didn't include the outgroups (potato, tomato and eggplant) in Figure 3, to clarify, we refer both Fig. S13 and Fig. 3 in the text, see line 227.

215-217

These results were already presented in lines 196-199. So rearrange these two sections.

The redundant sentence was removed.

233-235. From the supplementary methods it is not clear which sequence data were used for the GWAS. Please clarify. Indicate in the text how many accession were included in the GWAS

analysis.

We clarify the sequencing data, see line 239:

These resequencing data could be used for genome-wide association studies (GWAS).

We described the GWAS analysis in details in the supplementary information, see line 192- 200.

237 This likely explains...

> Clarify what you mean with “This”. Did the reads from chromosome 1 wrongly map to chr11? This could be confirmed if the chr11 reads that produced the high LOD score have low mapping quality

Rpi-amr4 on Chr01 belongs to the *Rpi-amr3* family, most *Rpi-amr3* homologs are in a same cluster on Chr11, that’s why some reads might be mapped to the *Rpi-amr3* locus on Chr11. We clarify this in the text, see Line 252-253.

242 The PITG_22825 responsive gene from SP2271 was mapped to the same position.

> Chr1 or chr 11? The RenSeq result that is shown in Fig S 13 is biased towards the candidate gene on Chr1. So provide evidence to support “the same position”

Same position on Chr01. We clarify this in the text. Here is the evidence:

1. For the GWAS analysis, we tried both SP2271 and SP1102 genomes as reference, and we got the same peak on Chr01.
2. For the BSA-RenSeq analysis, we mapped the RenSeq reads from two bulks (PITG_22825-R and PITG_22825-NR) to SP1102 and SP2271 genomes, and got same results.
3. We have cloned the *Rpi-amr4-2271* on Chr01 based on the SP2271 reference genome, although it’s auto-active in *N. benthamiana*, we observed a faster and stronger HR when co-expressed with PITG_22825 (Fig. 4c).
4. We generated *rpi-amr4* ko SP2271, the ko plants loss PITG_22825 responsiveness.

285 ... effector responsive genes ...

> the genes are not responsive to the effector. So consistently use effector response genes

We changed it to immune receptors.

314 ...BC1 population of SP2271(NR) x SP2300 (R) by transiently expressed

> replace by with and

Done

416-434. This section contains wild speculations about many different aspects that only weakly contribute to your conclusion that “non host” resistance can be achieved by R gene stacking. Please revise this section. Is it the sheer number of R genes in a stack that makes it unsurmountable by the simultaneous loss of effectors? Or are R genes from “non-host” plants more effective due to proven durability in co-evolution? Several reports about R gene stacking are available but are not referred to. Also realise that a gene can not be weak (as stated repeatedly). Probably you mean that Rpi gene mediated resistance is weak? Then please define weak. Be careful as HR amplitude (what you are studying in this manuscript mainly) does not necessarily correspond with disease resistance.

Thanks for the suggestions. We rewrote this paragraph, all the speculation sentences were removed, R gene stacking references were added.

439 Conceivably, Rpi-amr4, Rpi amr16 and Rpi-amr17 might also recognize the effectors from other important Phytophthora pathogens and confer resistance against multiple diseases. Ø This is just a loose remark. It must either be removed or substantiated to make a decent point.

We agree, this paragraph was removed. 497

8. Vossen, E. A. G. et al.

>van der Vossen Corrected

Suggestions to improve Tables and Figures Table

1: Title is missing

Added

It is not explained why anchor rate and chromosome number are missing for accession 2275

We only generated Hi-C data for SP1102, SP2271 and SP2273, see line 86-88. SP2275 was only assembled into contig level. Thus, the anchor rate and chromosome number were not provided in Table 1.

Figure 2:

Not all abbreviations/gene names in figure 2a and 2b are correct, explained, and/or referred to.

We have provided more information in the legend of Figure 2b.

Figure 2a and f: The PDF file provides insufficient resolution to view all information We provided high-resolution figure.

Figure 2b. Does the length of yellow and red line/dots proportional to the number of genes found?

Yes, each *NLR* gene was visualized by a small red/yellow square.

The position of the gene names should be indicated consistently to the left OR right side of the chromosome

All gene names were moved to the left.

For the comprehension and coherence with Fig 2e and 2f it would be good to include the position of the large *NRC4b*-like clade, *NRC5* and *NRC 6* as well

The position of *NRC4b* clade, *NRC5* and *NRC6* were marked in the updated Figure 2b.

Figure 2e the *S americanum* NRC orthologs should be named Sa instead of SP1102, to be consistent with the naming of the other species

The current name correlate with the genomic annotation of SP1102, and the manual annotation file in supplementary dataset 1-3. We think it would be handy for the readers to locate the gene based on their original name. We assigned some “Sa” names in Figure 2f.

Figure 2d. Indicate which color corresponds to manually curated or automatically predicted We added this information in the figure legend.

Figure 3: specify the transient expression system that was used We added this information in Table S3.

Figure 4: 4c What is HF? The text mentions Flag

It is a HisFlag tag, we replaced all HF or Flag with HisFlag.

4d: appears after figure 4f. So change order. The y axis values (lesion size?) must be added. Rpi-amr4_1102 must be mentioned in the legend.

We changed the order of figure 4d and figure 4f; The y axis values were added; Rpi-amr4- 1102 is mentioned in the legend.

Figure S2: The title should read: k-mer frequency distribution in *S. americanum* genome reads

Thanks for the comment, we have renamed the title to “K-mer frequency distribution based on Illumina reads of *S. americanum* genome”

Figure S4: indicate which BUSCO database has been used

We described the version of BUSCO database that has been used in this study in Methods section (line 37-39 in Supplementary Information). For convenience, we also added the sentence “The solanales_odb10 database was used for BUSCO evaluation.” to the figure legend.

Figure S5: indicate which window size was used

Each dot represents a pair-wise alignment, only alignments with length ≥ 100 bp and identity $\geq 80\%$ were kept. We also added this sentence to the figure legend.

Figure S6: indicate which reference genome was used

We have added the sentence “The genome of SP1102 was used as the reference.” to the figure legend and also added “Reference: SP1102” to the figure.

Figure S9b: The different box colors must be explained. The identity threshold for the

connecting lines should be given. Did you use a specific software tool to generate this image?

Clinker (<https://github.com/gamcil/clinker>) was used for clustering and the visualization. We have added the details in the figure legend, see Supplementary information line 510-512.

Figure S10: Explain abbreviations. Pp: Phytophthora palmivora? Why didn't you use Pi? Explain in the legend

PpAVRamr1 is the AVRamr1 homolog from *P. parasitica*, it gives stronger HR than the *P. infestans* T-30 allele (PITG_07569) when co-expressed with Rpi-amr1-2273, see Figure 3 in ¹. We added this information in the figure legend.

Figure S12: Specify a, b, and c

The legend is insufficiently describing the figure. Briefly describe the data and methods used for the phylogeny, the population structure (color scheme explanation), and inbreeding coefficient analysis.

We described more details in the figure legend. See line 520-534.

Figure S13: separate legends for b and

Provide a definition of informative SNPs.

mention the reference genome

improve the column headers in table C so that it is clear which frequency is meant. frequency of the Alt allele in NR bulk?

We have improved the legend of Figure S14 as suggested.

Reviewer #4:

Remarks to the Author:

This study has deployed wild accessions of *Solanum americanum* to clone novel immune receptors that recognize *Phytophthora infestans* RXLR effectors. *S. americanum* is resistant to *P. infestans*, yet can be used as source for engineering late blight (caused by *P. infestans*) resistance in potato genotypes used for production. Four *S. americanum* reference genomes were generated that were used to identify novel R gene specificities in 52 accessions, each challenged with 315 *P. infestans* effectors. This matrix search identified a total of three new R

gene specificities. The novelty and brilliance of this study lies in the use of a non-host plant to tap on new R genes, an approach that is much more efficient when diploid wild accessions are used rather than higher cultivated genotypes with often higher ploidy levels. I rate this a breakthrough study in R gene identification which, as the authors correctly state, could be applied to any crops and any diseases. Identification of three novel R genes provided a nice proof of principle. The genes identified behave, however, just as any other R genes (with limited novel insight into R gene function).

the grammar in sentences lines 55 and 254/255 is incorrect

We thank the reviewer for recognizing the novelty and significance of our work. And we agree that the current effectoromics pipeline can only be used for identifying the R genes of NLR type. However, with the genomic and genetic resources, this pipeline can be used for screening pathogen apoplastic effectors, and identification of plant surface immune receptors. Similarly, this work will lead the future discovery of atypical R genes or S genes in *S. americanum* by forward and reversed genetics.

We corrected the errors in line 55 and line 254-255.

Reference:

1. Witek, K. *et al.* A complex resistance locus in *Solanum americanum* recognizes a conserved *Phytophthora* effector. *Nature Plants* **7**, 198–208 (2021).
2. Panstruga, R. & Moscou, M. J. What is the Molecular Basis of Nonhost Resistance? *MPMI* **33**, 1253–1264 (2020).
3. Lin, X. *et al.* A potato late blight resistance gene protects against multiple *Phytophthora* species by recognizing a broadly conserved RXLR-WY effector. *Molecular Plant* 1–41 (2022).
4. Adachi, H. *et al.* An atypical NLR protein modulates the NRC immune receptor network. *BioRxiv* (2021).
5. Witek, K. *et al.* Accelerated cloning of a potato late blight–resistance gene using RenSeq and SMRT sequencing. *Nature Biotechnology* **34**, 656–660 (2016).
6. Rietman, H. *et al.* Qualitative and quantitative late blight resistance in the potato cultivar Sarpo Mira is determined by the perception of five distinct RXLR effectors. *MPMI* **25**, 910–919 (2012).
7. Derevnina, L. *et al.* Plant pathogens convergently evolved to counteract redundant nodes of an NLR immune receptor network. *PLOS Biology* **19**, e3001136 (2021).
8. Zhang, C. *et al.* Genome design of hybrid potato. *Cell* **184**, 3873–3883 (2021).

Decision Letter, first revision:

9th Feb 2023

Dear Professor Jones,

Your Article, "Solanum americanum genome-assisted discovery of new immune receptors that detect potato late blight effectors" has now been seen by 3 referees. You will see from their comments below that while they find your work of interest, some important points are raised by reviewer #3. We are interested in the possibility of publishing your study in Nature Genetics, but would like to consider your response to these concerns in the form of a revised manuscript before we make a final decision on publication.

We therefore invite you to revise your manuscript taking into account all reviewer and editor comments. Please highlight all changes in the manuscript text file. At this stage we will need you to upload a copy of the manuscript in MS Word .docx or similar editable format.

*2) If you have not done so already please begin to revise your manuscript so that it conforms to our Article format instructions, available [here](http://www.nature.com/ng/authors/article_types/index.html). Refer also to any guidelines provided in this letter.

[redacted]

Note: This URL links to your confidential home page and associated information

about manuscripts you may have submitted, or that you are reviewing for us. If you wish to forward this email to co-authors, please delete the link to your homepage.

We hope to receive your revised manuscript within four to eight weeks. If you cannot send it within this time, please let us know.

Sincerely,
Wei

Wei Li, PhD
Senior Editor
Nature Genetics
New York, NY 10004, USA
www.nature.com/ng

Reviewers' Comments:

Reviewer #1:

Remarks to the Author:

Most of the concerns that I raised in earlier review process have been resolved in the revised version of manuscript. Thus, I think the manuscript is acceptable as current form.

Reviewer #2:

Remarks to the Author:

The revised manuscript clarified certain key points of the study. I agree that the defense/counter-defense network likely contributes to the weak disease resistance phenotype in the knock out/overexpression plants related to the newly identified effector-NLR pair.

Reviewer #3:

Remarks to the Author:

All my comments from the previous review have been adequately addressed.

- The addition of complementation of Rpi-amr4 in stable transgenic lines of bentha convincingly show this is a genuine Rpi gene. The renaming of the effector receptors and the addition of the transient complementation data take away the concerns about this part of the research. Initially I found the name "immune receptor", for a receptor that does not induce immunity a bit confusing. After consulting literature, especially animal studies, it came clear to me that just the perception of a component leading to any response from the immune system is referred to as an immune receptor.
- Rpi gene nomenclature is now much more consistent and approaches a consensus among labs.

Unfortunately, new additions and changes to this version of the manuscript raise new questions:

Title: *Solanum americanum* genome-assisted discovery of new immune receptors that detect potato late blight effectors

> I have a problem with "late blight effectors". In the narrow definition of pathogen effectors, you describe *P infestans* effectors. In a broader definition these proteins could be late blight effectors if their role in late blight development is shown. This is not the case. So a revision of the title is needed.

Line 64: "potato genome design".

> As far as I understand the definition of "genome design" from the cited work by some co-authors on this paper it means the design of genomes for cross-breeding. The authors agree that it is not feasible to introduce the Rpi-amr genes into potato germplasm through cross breeding. Maybe I misunderstood the definition of genome design but then it must be better defined what it means. Anyway, also for the purpose of informing legislators, it must be made clear that Novel Plant Breeding Techniques are needed to deploy the currently described resources.

Line 77: "potato"

> too broad description of the Desiree plants in Fig S1. There are numerous potato varieties that are more resistant to T30-4 than SP2271.

Line 142: might BE associated

Line 164:

> check grammar.

> This new section is meant to illustrate the completeness of NLR-ome coverage as performed in this study. However, a few SNPs, deletion or insertion of a LRR can result in a different recognition specificity. Also it is strange to apply the core vs dispensable nomenclature in the NLR context. So, is orthofinder the most suitable tool to assess the completeness of the NLR-ome?

Line 194: investigation OF NLR gene evolution

Line 204: "upon infection of potato"

> Ambiguous description. Better: During colonisation of a susceptible potato leaves

Line 329: Rpi gene.

> Replace with Immune receptor

Line 360: the late blight resistance of R02860 and R04373...

> Genes can not be resistant to late blight

Line 365: These results indicate that although R02860 and R04373 can recognize the RXLR effectors PITG_02860 and PITG_04373 from *P. infestans*, they can be overcome by *P. infestans*.

> A gene can not be overcome.

Line 354: however the HR phenotype was not restored in these knockout lines (Supplementary Fig. 18e).

> You must address this unexpected result, at least in the discussion. Maybe even simple experimental additions could be done. Could Rpi-amr1 or Rpi-amr3 be co-receptors? Is a truncated R04373 gene producing interfering RNA? The first hypothesis could be tested by co-infiltration in a non-responsive plant that lack/contain Rpi-amr1 or 3. The latter hypothesis could be tested by co-infiltration of a silencing suppressor.

Supplementary Figure 17: Rpi-amr16 must be updated

Author Rebuttal, first revision:

Please see below for our responses to reviewer 3's comments.

Title: *Solanum americanum* genome-assisted discovery of new immune receptors that detect potato late blight effectors

> I have a problem with "late blight effectors". In the narrow definition of pathogen effectors, you describe *P. infestans* effectors. In a broader definition these proteins could be late blight effectors if their role in late blight development is shown. This is not the case. So a revision of the title is needed.

We are sorry that Reviewer 3, but not 1 or 2, has a problem with this title. For the MPMI community this language is clear and well justified. NLR proteins, often but not always encoded by Resistance genes are immune receptors that detect specific pathogen effectors (predicted for *P. infestans* by the presence of a signal peptide and an RxLR motif) and activate defence. Using the Sam genome, we discovered new immune receptors that detect *P. infestans* RxLR effectors.

Line 64: "potato genome design".

> As far as I understand the definition of "genome design" from the cited work by some co-authors on this paper it means the design of genomes for cross-breeding. The authors agree that it is not feasible to introduce the Rpi-amr genes into potato germplasm through cross breeding. Maybe I misunderstood the definition of genome design but then it must be better

defined what it means. Anyway, also for the purpose of informing legislators, it must be made clear that Novel Plant Breeding Techniques are needed to deploy the currently described resources.

Thanks for the suggestions, we rephrased the sentence in line 64-65: "In combination with potato genome design driven by potato genomics taking advantage of novel plant breeding technologies"

Line 77: "potato"

> too broad description of the Desiree plants in Fig S1. There are numerous potato varieties that are more resistant to T30-4 than SP2271.

True- some varieties are more resistant than SP2271 eg var Toluca. We will make text more specific, we added cv. Desiree in line 78.

Line 142: might BE associated

Corrected

Line 164:

> check grammar.

Thanks for pointing that out, we have rephrased the sentence 'suggesting the representative of *S. americanum* NLR repertoire.' to 'which suggests that the accessions in our research are representative of *S. americanum* NLR repertoires.'

> This new section is meant to illustrate the completeness of NLR-ome coverage as performed in this study. However, a few SNPs, deletion or insertion of a LRR can result in a different recognition specificity. Also it is strange to apply the core vs dispensable nomenclature in the NLR context. So, is orthofinder the most suitable tool to assess the completeness of the NLR-ome?

Rev3 is correct in that even core genes can show variation that affects recognition capacity. Additionally, SNPs, deletions or insertions can also lead to functional variation in non-NLR genes. Thus, the analysis approaches for the pan-genome and pan-NLRome are similar. In the current revision, we have included two analyses to describe the *S. americanum* NLRome.

1. Figure 2a, this figure reflects the P/A polymorphism and identity of individual NLRs from different Sam accessions based on the SP1102 reference genome;

2. The new Pan-NLRome analysis that was suggested by Reviewer 1 (Figure S9). Here we aim to present an overview of the *S. americanum* NLRome. The method was improved from the pan-NLRome research in *Arabidopsis*. In this study, OrthoFinder was deployed to perform all-vs-all alignments, adjust the alignment bias and classify orthogroups with the embedded MCL algorithm; these steps were performed by separate software in the former study. OrthoFinder is a widely used tool designed for comparative genomics analysis with high accuracy and performance. The nomenclature of pan-genome or pan-NLRome analysis is adopted from the original research in bacteria and modified depending on the research context. In our research, the original nomenclature system, 'core' and 'dispensable', were used and we also introduced the 'unique' category to describe NLRs that were present in only one accession.
Line 194: investigation OF NLR gene evolution

Corrected.

Line 204: “upon infection of potato”

> Ambiguous description. Better: During colonisation of a susceptible potato leaves

Adopted as

“During colonisation of a susceptible potato leaf “

Line 329: Rpi gene.

> Replace with Immune receptor

Adopted

Line 360: the late blight resistance of R02860 and R04373...

> Genes can not be resistant to late blight

Correct- we will adjust to “the late blight resistance CONFERRED BY R02860 and R04373

Line 365: These results indicate that although R02860 and R04373 can recognize the RXLR effectors PITG_02860 and PITG_04373 from *P. infestans*, they can be overcome by *P. infestans*.

> A gene can not be overcome.

We adjusted to “the resistance conferred by them can be overcome by *P. infestans*.”

Line 354: however the HR phenotype was not restored in these knockout lines (Supplementary Fig. 18e).

> You must address this unexpected result, at least in the discussion. Maybe even simple experimental additions could be done. Could Rpi-amr1 or Rpi-amr3 be co-receptors? Is a truncated R04373 gene producing interfering RNA? The first hypothesis could be tested by co-infiltration in a non-responsive plant that lack/contain Rpi-amr1 or 3. The latter hypothesis could be tested by co-infiltration of a silencing suppressor.

Thanks to reviewer's insightful suggestions, but we don't think Rpi-amr1 and Rpi-amr3 can be co-receptors, because we have verified the functionality of R04373 in *N. benthamiana*, which lacks Rpi-amr1 and Rpi-amr3. For hypothesis 2, we can acknowledge this as a possibility and we added this suggestion in the discussion. "we hypothesized that truncated R04373 paralogs might produce interfering RNAs"

Supplementary Figure 17: Rpi-amr16 must be updated

We thank the reviewer for his/her sharp eyes, we have corrected Supplementary Figure 17.

Sincerely yours,

Prof Jonathan D G Jones FRS

Decision Letter, second revision:

24th Apr 2023

Dear Dr. Jones,

Thank you for submitting your revised manuscript "Solanum americanum genome-assisted discovery of new immune receptors that detect potato late blight effectors" (NG-A60796R1). It has now been seen by the original referees. The reviewers find that the paper has improved in revision, and therefore we'll be happy in principle to publish it in Nature Genetics, pending minor revisions to comply with our editorial and formatting guidelines.

Sincerely,
Wei

Wei Li, PhD
Senior Editor
Nature Genetics
New York, NY 10004, USA
www.nature.com/ng

Final Decision Letter:

21st Jul 2023

Dear Dr. Jones,

I am delighted to say that your manuscript "Solanum americanum genome-assisted discovery of immune receptors that detect potato late blight pathogen effectors" has been accepted for publication in an upcoming issue of Nature Genetics.

Your paper will be published online after we receive your corrections and will appear in print in the next available issue. You can find out your date of online publication by contacting the Nature Press Office (press@nature.com) after sending your e-proof corrections. Now is the time to inform your Public Relations or Press Office about your paper, as they might be interested in promoting its publication. This will allow them time to prepare an accurate and satisfactory press release. Include your manuscript tracking number (NG-A60796R2) and the name of the journal, which they will need when they contact our Press Office.

Before your paper is published online, we shall be distributing a press release to news organizations worldwide, which may very well include details of your work. We are happy for your institution or

funding agency to prepare its own press release, but it must mention the embargo date and Nature Genetics. Our Press Office may contact you closer to the time of publication, but if you or your Press Office have any enquiries in the meantime, please contact press@nature.com.

Please note that *Nature Genetics* is a Transformative Journal (TJ). Authors may publish their research with us through the traditional subscription access route or make their paper immediately open access through payment of an article-processing charge (APC). Authors will not be required to make a final decision about access to their article until it has been accepted. [Find out more about Transformative Journals](https://www.springernature.com/gp/open-research/transformative-journals)

Authors may need to take specific actions to achieve [compliance with funder and institutional open access mandates](https://www.springernature.com/gp/open-research/funding/policy-compliance-faqs). If your research is supported by a funder that requires immediate open access (e.g. according to [Plan S principles](https://www.springernature.com/gp/open-research/plan-s-compliance)) then you should select the gold OA route, and we will direct you to the compliant route where possible. For authors selecting the subscription publication route, the journal's standard licensing terms will need to be accepted, including [self-archiving-and-license-to-publish](https://www.nature.com/nature-portfolio/editorial-policies/self-archiving-and-license-to-publish). Those licensing terms will supersede any other terms that the author or any third party may assert apply to any version of the manuscript.

Please note that Nature Portfolio offers an immediate open access option only for papers that were first submitted after 1 January, 2021.

If you have not already done so, we invite you to upload the step-by-step protocols used in this manuscript to the Protocols Exchange, part of our on-line web resource, natureprotocols.com. If you complete the upload by the time you receive your manuscript proofs, we can insert links in your article that lead directly to the protocol details. Your protocol will be made freely available upon publication of your paper. By participating in natureprotocols.com, you are enabling researchers to more readily reproduce or adapt the methodology you use. [Natureprotocols.com](https://natureprotocols.com) is fully searchable, providing your protocols and paper with increased utility and visibility. Please submit your protocol to <https://protocolexchange.researchsquare.com/>. After entering your [nature.com](https://www.nature.com) username and password you will need to enter your manuscript number (NG-A60796R2). Further information can be found at <https://www.nature.com/nature-portfolio/editorial-policies/reporting-standards#protocols>

Sincerely,
Wei

Wei Li, PhD
Senior Editor
Nature Genetics
New York, NY 10004, USA
www.nature.com/ng